# Agile perching maneuvers in birds and morphing-wing drones

Valentin Wüest [1] ✉, Simon Jeger[1], Mir Feroskhan [2], Enrico Ajanic[1], Fabio Bergonti [3] & Dario Floreano [1] ✉

Avian perching maneuvers are one of the most frequent and agile flight scenarios, where highly optimized flight trajectories, produced by rapid wing and tail morphing that generate high angular rates and accelerations, reduce kinetic energy at impact. While the behavioral, anatomical, and aerodynamic factors involved in these maneuvers are well described, the underlying control strategies are poorly understood. Here, we use optimal control methods on an avian-inspired drone with morphing wing and tail to test a recent hypothesis derived from perching maneuver experiments of Harris' hawks that birds minimize the distance flown at high angles of attack to dissipate kinetic energy before impact. The resulting drone flight trajectories, morphing sequence, and kinetic energy distribution resemble those measured in birds. Furthermore, experimental manipulation of the wings that would be difficult or unethical with animals reveals the morphing factors that are critical for optimal perching maneuver performance of birds and morphing-wing drones.

Birds of prey are capable of agile maneuvers that are unmatched by winged drones of similar size and mass[1,2]. They can dynamically adjust the sweep, area, and inclination of their wings and tail to adapt to momentary aerodynamic demands[3,4]. Researchers have conducted anatomical, aerodynamic, and behavioral observations to better understand the underpinning morphological and control strategies.

Anatomical studies have shown that the majority of wing shape alterations are achieved by coupled actuation of elbow and wrist joints in the wing[5], which results in variations of wing sweep that allows the bird to both shift and enlarge the lifting surface. Wind tunnel experiments on gull wings and aerodynamic modeling of peregrine falcons during pullout maneuvers revealed that these birds employ sweeping motion to shift the aerodynamic center forward. This change reduces static pitch stability, which is desirable for agile flight because the effects of control actions are magnified[6,7]. Furthermore, sweeping the wings enlarges the lifting surface and in turn increases the aerodynamic forces and moments[8,9], allowing birds to rapidly change their orientation and flight direction. This ability to rapidly change orientation and flight direction epitomizes agility, defined in our context as the ability to rapidly change linear and angular velocities[9]. Both wing

and tail contribute to the bird's overall lift[10,11]. In slow, aggressive flight, birds spread their tail to produce substantial lift and drag forces to maneuver their flight path[4,12]. In addition to producing pitch moments with their wings, birds also tilt their tail upwards to increase the positive pitching moment and change orientation more rapidly[12]; however, this action reduces lift forces. To balance the costs and benefits of such actions, birds display complex control strategies to swiftly transition between diverse morphological configurations to meet flight demands[13].

Understanding avian control strategies requires the examination of dynamic flight trajectories. Field studies of birds in free flight can provide useful insights into the rapid morphological changes that underpin agile flight maneuvers. For instance, observations of the perching maneuver by the Steppe eagle revealed that the bird starts with a fast pitch-up action by raising the tail and sweeping the wings forward, allowing it to reach a high angle of attack in less than 0.2 s. At this point, it fully extends its wings and tail to reduce speed and finally tilts the tail downwards to stop pitch rotation[13] and increase decelerating forces. Such studies, however, offer a limited understanding of avian control objectives and are constrained by observational

[1]Laboratory of Intelligent Systems, École Polytechnique Fédérale de Lausanne, Lausanne, Vaud, Switzerland. [2]School of Mechanical and Aerospace Engineering, Nanyang Technological University, Singapore, Singapore, Singapore. [3]Artificial and Mechanical Intelligence Laboratory, Istituto Italiano di Tecnologia, Genova, Genova, Italy. ✉e-mail: valentinwuest@gmail.com; dario.floreano@epfl.ch

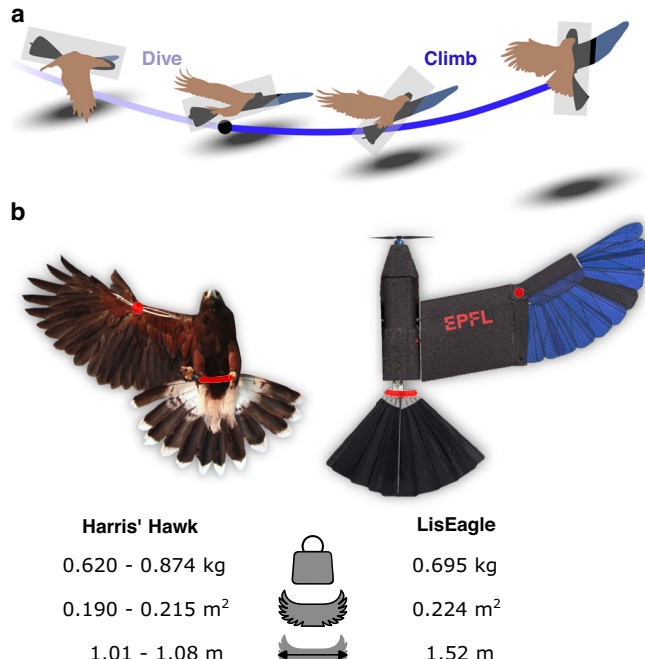

**Fig. 1 | The perching maneuver by Harris' hawks and avian-inspired drone.**
**a** Schematic overview of a typical perching maneuver consisting of a dive phase (light blue line) and of an agile climb phase (blue line) (see also Supplementary Movie 1, adapted from ref. [26]); the illustrated bird shows the optimal control strategies displayed by the Harris' hawk (light brown) and by the avian-inspired drone (black/blue). **b** Comparison of Harris' hawks of the study[26] and the avian-inspired LisEagle drone[17]. We indicate the wing and tail sweep joints in red and list the mass, wing area, and wing span of the two.

uncertainties and other factors that may affect their natural behavior in the field[1].

Avian-inspired robots can serve as models to systematically investigate the three factors of anatomical and kinematic motion ability of wing and tail morphing, their aerodynamic impact on static and dynamic flight conditions, and the optimal control strategies under a variety of well-defined flight behaviors[1,9,14–20]. For example, a drone with partly folding wings made of artificial feathers was used to show increased maneuverability via wing expansion and reduced drag via wing folding[16], similarly to birds[21]. Additionally, researchers added a sweeping mechanism to a fixed-wing drone to show that it could pitch up rapidly to reduce its flight speed[15]. However, the drone did not incorporate wing area changes as seen on birds, the influence of the sweeping on the maneuver was not studied, and no analogies to birds were drawn. Studies with avian-inspired drones revealed the aerodynamic impact of synergistic wing and tail morphing on agility, maneuverability, pitch stability, and energetic efficiency at diverse flight speeds[9]. It was shown that wing twisting is more effective for rolling than asymmetric wing morphing when flying at moderate angles of attack, typical of cruising flight[17]. Research on a feathered robotic wing shed light on the wing morphing of pigeons to explain how underactuated wing-sweep kinematics and passively interlocking feathers[18] enable rapid and robust sweeping to initiate turning maneuvers[19]. Additionally, a biomimetic feathered robotic jackdaw wing was employed in wind tunnel experiments, exploring a range of flapping and morphing patterns and frequencies to reveal the energetic benefits in force generation these birds achieve by folding their wings during the upstroke[20].

Birds exhibit complex control strategies that require the consideration of the interplay of the three previously mentioned factors to balance costs and benefits by swiftly changing morphological configurations. The drones described above, on the other hand, have been steered through comparatively simple maneuvers by either manual teleoperation[9,16,17,19] or reactive scheduled Proportional-Integral controllers[22]. To fully investigate and leverage the potential of morphing wing and tail during more complex and agile avian-like flight maneuvers, it becomes crucial to achieve high angular acceleration and high aerodynamic forces[4]. However, these demands require opposing actuator inputs. For example, producing high angular acceleration would demand the tail to be inclined upwards, whereas producing a high lift force would require it to be deflected downwards[9]. Therefore, an understanding of the underlying control objective and foresight into upcoming aerodynamic conditions are important to reconcile these conflicting aerodynamic needs. The importance of such anticipatory control strategies becomes even more pronounced when operating near the robot's actuation limits where the actuation bandwidth is severely limited[23,24].

In this context, the perching maneuver displayed by birds of prey is a remarkable example of an agile flight maneuver, which is characterized by a dive phase and a climb phase, demanding flight across a diverse aerodynamic spectrum to achieve and balance high angular acceleration and high aerodynamic forces[25]. The perching maneuver allows the birds to dissipate kinetic energy within a short distance in preparation for the impact and grasp[13]. A recent laboratory study[26] of Harris hawks' engaged in perching maneuvers revealed that birds initiate the maneuver with a powered dive that slightly increases kinetic energy (light blue line in Fig. 1a). Once the lowest point is reached, birds fly upwards towards the perching target in an unpowered climb (blue line in Fig. 1a). In this agile phase of the maneuver, they swiftly modify wing sweep, tail sweep, and tail incidence to transition from gliding flight to high-angle-of-attack flight for energy dissipation and a soft landing[13]. The authors hypothesized that besides reducing the impact energy, the bird's control strategy aims at minimizing the distance flown at high angle of attack, where the bird has little control authority, rather than minimizing either flight time or energy consumed for propulsion. Birds commonly do this for migration and commuting[27] and this is also done in robotic implementations of autonomous perching maneuver control[15,28].

Here, we test this hypothesis using optimization methods in aerodynamically grounded simulations of an avian-inspired drone with morphing wing and tail (Fig. 2) to study the dive and climb phase of the perching maneuver (excluding the physical grasping). Specifically, we compute flight trajectories to obtain control strategies for an avian-inspired drone minimizing the objective of impact energy and distance flown at high angle of attack in experimental conditions similar to those used with birds of prey[13,26]. The simulation results indicate that the optimal flight path and aerodynamic key characteristics of the flown horizontal distance, as well as relative kinetic energy at impact, match those measured in the Harris' hawks. Systematically limiting the range of wing and tail sweep led to a deterioration in aerodynamic performance, emphasizing the importance of the wing kinematics and the anatomical ability to morph. Furthermore, we show that the morphing control strategy that leads to these optimal trajectories matches qualitative observations of wing and tail morphing reported in the literature on perching maneuvers of birds of prey[12,13]. Finally, we validate the optimal control strategy on the physical drone and show that its flight path and kinematic morphing actuation closely match the video footage of Harris' hawks measured in laboratory conditions[26]. Systematically limiting the range of wing and tail sweep led to a deterioration in aerodynamic performance, emphasizing the importance of the wing kinematics and the anatomical ability to morph. These results corroborate the hypothesis that birds of prey develop optimal perching maneuver strategies to reduce kinetic energy and minimize flight duration at unstable flight conditions and show how quantitative descriptions of bird behaviors can be used to derive optimal control strategies to both study avian flight and achieve avian agility on non-biological fliers.

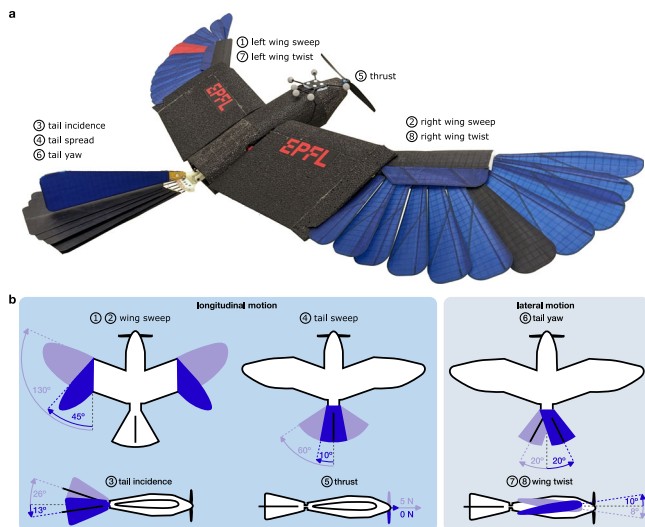

**Fig. 2 | Avian-inspired drone. a** The avian-inspired drone, LisEagle, used in this work. **b** Illustration of its eight degrees of freedom. This study focuses on longitudinal motion, which is affected by symmetric sweeping of left and right wing, tail incidence, tail sweep, and thrust. The control of these degrees of freedom is given to the trajectory optimization method. Instead, lateral displacements induced by mechanical asymmetries of the drone feathers and actuator responses are stabilized by reactive Proportional-Derivative (PD) controllers where tail yaw correct yaw motion and asymmetric left and right wing twist correct roll motion.

## Results

The drone used in this study (code-named LisEagle, Figs. 1b, 2a) is endowed with feathered wings and tail with controllable sweep and incidence angles observed in birds of prey during agile maneuvers, and a frontal propeller to generate thrust[9,17]. The drone's weight (695 g) falls within the range of the four Harris' hawks (620–874 g) observed in the animal study, and its nominal wing area (0.224 m²) is only slightly larger than that of the birds (0.190–0.210 m²).

For the sake of simplicity, we study optimal control of the aerial surfaces along the vertical plane that connects the starting and target points. The drone can use the same wing and tail motion observed in birds during rapid pitch-up maneuvers[1,13], namely wing sweep, tail sweep, and tail incidence (Fig. 2a, b, Supplementary Movie 2), to control its longitudinal motion. Wing and tail sweeping increases lifting surfaces, thus producing larger aerodynamic moments and forces. Sweeping the wings forward also changes the relative position of the center of lift with respect to the center of gravity[3] (collapsed: −1.4 cm, extended: 7.2 cm, see Supplementary Table 1, Supplementary Note 9), thus reducing longitudinal stability and increasing agility. These two aerodynamic effects of wing and tail sweeping enable the drone to rapidly rotate upwards and generate large aerodynamic forces[20]. While birds of prey primarily utilize flapping for thrust generation during the powered dive phase prior to perching[26], in this study we use a propeller to generate thrust and focus on identifying wing and tail control strategies during the agile, unpowered climb phase when large birds tend not to flap[26]. When validating the optimal perching maneuver control strategies on the real drone, we employ gain scheduled Proportional-Derivative (PD) control of the tail yaw angle and asymmetric actuation of the wing twist (Fig. 2a, b, Supplementary Movie 2) to laterally stabilize the drone in terms of yaw and roll respectively (details in "Methods").

We derived an aerodynamically grounded model for our drone using wind tunnel measurements at varying wind speeds, angles of attack, and wing and tail configurations (see "Methods") and also took actuator limitations into account. This model enabled us to conduct trajectory optimization experiments of the agile climb phase of a perching maneuver in simulation. During this phase, a bird starts from

a glide and dissipates significant energy before reaching its target point[26]. We recreate a similar scenario to obtain trajectories that allow a comparative assessment of the resulting control strategy. To compare the performance of our results to the Harris' hawk, we evaluate the perching maneuver performance in terms of relative impact energy (a measure of kinetic energy reduction to the means of reduced impact) and required distance (a measure to compare flight path and space requirements). We further qualitatively compare the wing and tail actuation sequence leading to the trajectory with the wing and tail motion observed in birds[12,13,26].

We used trajectory optimization criteria identified in studies of similarly sized birds of prey engaged in the perching maneuver[12,13,26] with the goal of testing the hypothesis that birds minimize the distance flown at high angle of attack and of studying the role of morphing strategies in perching maneuver trajectories and performance. For example, Harris' hawks were permitted to fly from a start point to a perch point positioned at an equal elevation at distances between 5 and 12 m[26]. These birds commenced their flight with a powered dive. Similarly, Steppe eagles flying in their natural habitat[13] were reported to display a diving behavior to a depth of 1 m below their perching target. This behavior was also described in Harris' hawk experiments with a 12 m distance between start and perch points[26]. Consequently, in our trajectory optimization experiments, we considered a range of distances centered around 12 m (9–15 m in 1 m increments), initialized the drone in a straight flight condition at the nominal speed of the drone of 10 ms⁻¹, and limited the maximum downward movement to 1 m, as observed in birds.

In summary, the trajectory optimization is formulated as a non-linear program[29], characterized by the dynamics of our avian-inspired drone, where we minimize the impact energy and distance flown at high angle of attack with the constraint of an initial horizontal distance between 9 and 15 m, an initial forward velocity of 10 ms⁻¹, and a maximum downward vertical distance of 1 m, while upward movement remains unrestricted (Fig. 3a). As a control experiment, we also performed two sets of trajectory optimization experiments where we minimize the total flight time and required propulsion energy, respectively, instead of the distance flown at high angles of attack. To solve the optimization problem, we employ the Interior Point OPTimizer (Ipopt) method[30] (further details in Methods).

### Optimal perching trajectories display dive and climb

The optimal flight trajectories showcase a distinct and consistent swooping pattern, which can be broken down into two main phases: the dive and the climb phase (see Supplementary Movie 3). These are separated by the transition state, marked by the lowest point of the trajectory, as illustrated in (Fig. 3a, b)[26]. During the dive phase, the drone engages in a powered glide, descending with slight acceleration. This maneuver takes place within the allowed vertical space of 1 m, where its wings are straight and the tail is semi-extended and angled slightly upward. The climb phase represents the more agile part of the maneuver. Here, the drone performs an unpowered climb towards the target, while it pitches upwards to increase its angle of attack and rapidly reduces speed (Fig. 3b).

### Optimal perching maneuvers redistribute kinetic energy prior to impact

In the short distance covered during the climb phase, birds dissipate most kinetic energy to minimize the impact that must be absorbed to stop their motion[13,31]. Consequently, we analyzed the impact energy, i.e., the remaining kinetic energy, of optimal drone trajectories during the climb phase and compared them to those of birds. The bird's posture and ability to extend its legs forward enable it to touch the perching pole before the body reaches the pole. We estimated this effective distance at 0.5 m by analyzing video footage of Harris' hawk experiments (detailed in Supplementary Fig. 1, Supplementary Note 1).

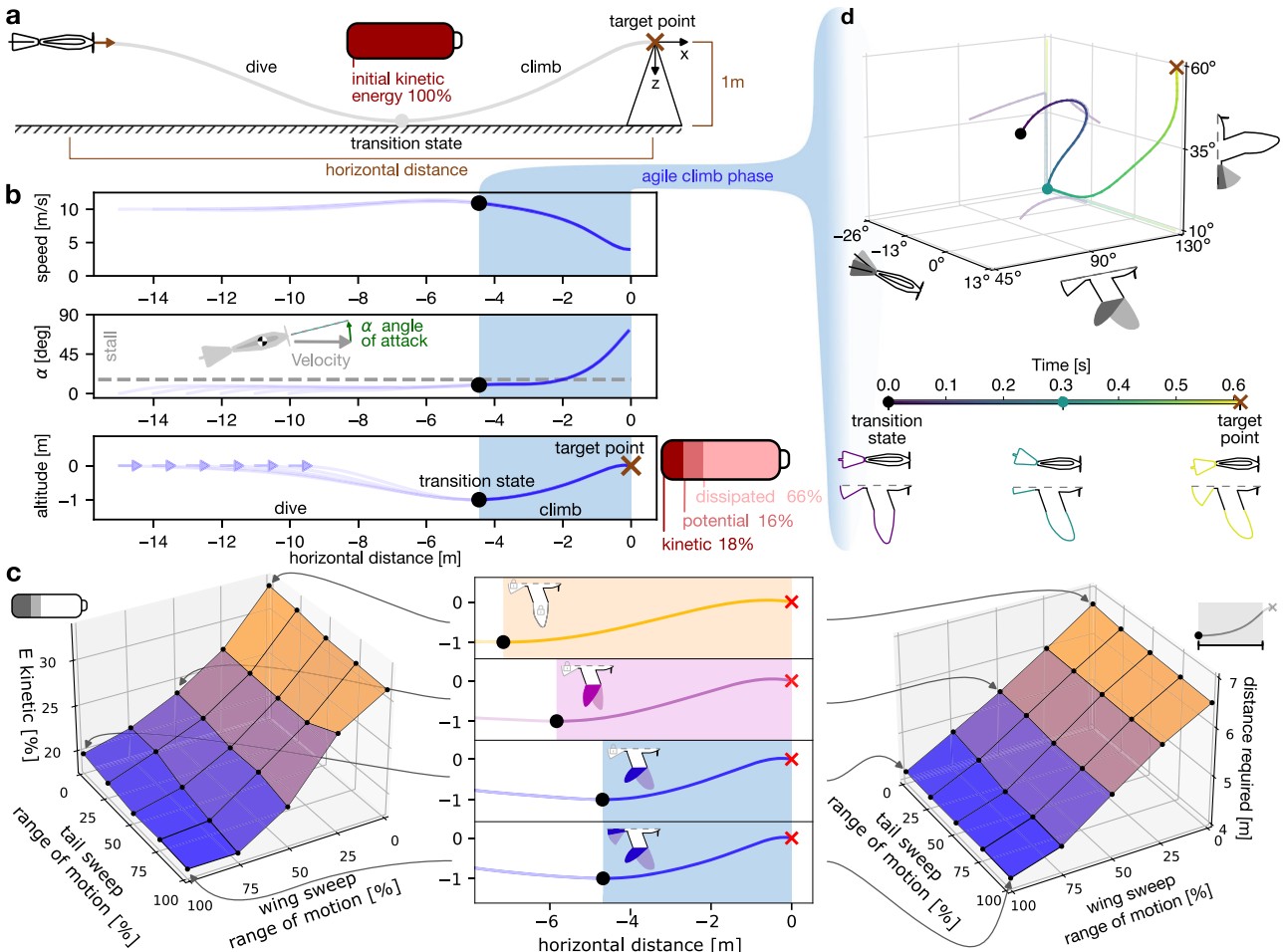

**Fig. 3 | Perching optimization in simulation. a** Visualization of the simulation experiment constraints. The drone is initialized in a straight flight condition at a velocity (brown arrow) of 10 ms⁻¹ and varying distance from the target point (brown cross). The drone is required to reach the target point and is allowed to use up to 1 m of space downward, while its motion upwards is not restricted. The transition state is defined at the lowest point of the trajectory, where all the energy is kinetic. **b** Perching flight trajectories that result from the optimization algorithm with the objective of minimizing distance flown at high angle of attack[26]. Starting points are in the range of 9–15 m, spaced in intervals of 1 m and are shown by an arrow. The climb phase is indicated by the shaded area, on the right of which, we show kinetic, potential, and dissipated energy at the target point. **c** Effect of a limited range of motion in wing and tail sweep on flight path and key characteristic metrics. The central panel shows the resulting flight paths, the left panel (E kinetic) displays the relative kinetic energy on impact, and the right panel (distance required) indicates the horizontal distance required for the perching maneuver (both surfaces linearly interpolated between experiments, indicated by black dots). **d** Drone trajectory during the climb phase in the morphological space defined by tail incidence, wing sweep, and tail sweep. The prominent 3-dimensional line shows the actuation sequence, with color change indicating time flow. The lighter lines depict the 2-dimensional projections on each parameter plane. The time scale at the bottom highlights the three distinct configurations of the drone during the climb phase. Source data are provided as a Source Data file.

Considering the absence of leg extension in the drone, energy measurements were conducted when the drone reached a distance of 0.5 m from the perching target. Lastly, we excluded trajectories that do not utilize the full 1-meter vertical space, as these would not accurately reflect the necessary horizontal distance required for an optimal perching maneuver or the comparative energy redistribution.

Starting from the maximum kinetic energy at the transition state (defined by the lowest point in the dive), we measure the remaining kinetic energy at impact, the potential energy gained by increasing altitude, and the dissipated energy. The flights from the Harris' hawk study reveal that when birds impact the perching point after a 4.2 m distance from the transition state, they dissipated 54% of the initial kinetic energy, converted 32% into potential energy, and remained with only 14% kinetic energy. Similarly, the simulated drone covered 4.1 m and dissipated 66% of the initial kinetic energy, converted 16% into potential energy, and remained with only 18% kinetic energy (Fig. 3b).

In contrast, the optimal flight trajectories found in control experiments with minimization of total flight time or required

propulsion energy display a distinctly different path where the transition to the climbing phase occurs later (3.0 m and 3.1 m from the perching point, respectively) (see Supplementary Fig. 2 Supplementary Note 2 for details).

## Wing and tail morphing results in higher kinetic energy dissipation

To assess the role of the morphing wing and tail in the perching maneuver, we gradually restricted the range of motion of wing and tail sweep to their center position (similar to the configuration at the transition state) and applied the trajectory optimization method in the same experimental conditions of the morphing-wing drone. In the condition where wing and tail sweep motions are blocked, only the propeller thrust and tail incidence can control longitudinal motion. This limited configuration most closely resembles that of a fixed-wing drone (Fig. 3c, orange trajectory) and serves as a control condition to understand the contribution of wing and tail sweep to the perching maneuver. Under those fixed-wing conditions, the drone covered a longer distance (6.6 m) during the climb phase than the morphing-

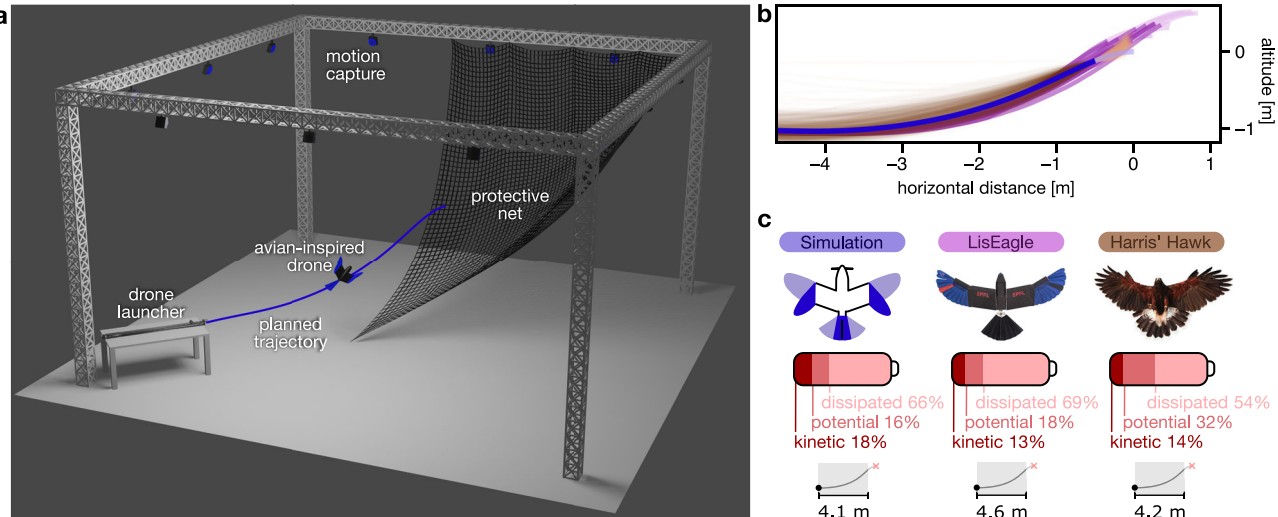

**Fig. 4 | Validation on drone. a** Experimental setup consisting of a launcher accelerating the drone to the transition state to fly along the planned trajectory while being tracked by a motion-tracking system and eventually landing in a protective net for recovery. **b** The planned optimal trajectory, the trajectories from the 12 LisEagle flights, and the 537 Harris' Hawks flights[26]. We align the optimal trajectory by aligning the point of impact of the Harris' Hawk with the corresponding point of the optimized drone trajectory. Beyond the impact point, each of the trajectories is shown in a faded color. **c** Energy ratios at impact of the simulated drone, the real drone, and Harris' hawks. Source data are provided as a Source Data file.

wing drone (4.1 m) (Fig. 3c, b). Notably, despite this increased distance, the fixed-wing model dissipated only 54% of its kinetic energy compared to 66% of energy dissipated by the morphing-wing drone. As a result, the kinetic energy 0.5 m before the target point was higher in the fixed-wing model and lead to an increase from 18% to 33% compared to the configuration where area morphing is allowed. To dissect the impact of the tail and wing adjustments separately, we analyzed how the required climbing distance and residual kinetic energy varied when their motion range was incrementally restricted (Fig. 3c). Both limitations raised the distance needed and the kinetic energy upon impact, but wing sweep lead to a much more pronounced effect than limitations in tail sweep.

## Wing morphing sequence matches bird's behavior
Next, we asked whether the wing and tail actuation strategies used in the optimized drone trajectories correspond to those displayed by birds of prey. Studies on the climb phase of perching maneuvers in birds, notably in Steppe eagles[13] and white backed vultures[12], revealed transitions through three distinct configurations. Video footage of Harris' hawks in laboratory settings displayed a similar actuation sequence[26]. Firstly, the bird glides with extended wings and a straight tail. Secondly, it sweeps the wings forward and elevates its tail. Finally, it flies at a high angle of attack with forward-swept wings and a downward inclined, fully extended tail.

The actuation sequence of the drone (Fig. 3d) performing the optimal trajectories described above displays a striking resemblance to that observed in the three bird species (see Supplementary Movie 4). The drone starts at the transition state with the wing extended at its middle angle between tucked and fully extended and the medium extended tail at a straight angle. It then sweeps the wings forward and increases the tail incidence angle (although, differently from birds, it reduces the tail size, which in our simulation leads to a higher pitch-up moment of 0.07 Nm, than an extended tail −0.55 Nm at this stage of the maneuver), producing a high pitch acceleration that increases the drone's angle of attack. Finally, when approaching the perching point, the drone keeps the wings swept forward and extends and lowers the tail similarly to birds. This final configuration results in high lift and drag forces that act similar to a cross-parachute to decelerate the drone[13].

## Validation on the physical drone
Lastly, we validated the optimal control strategy of the climb phase on the real LisEagle drone. For sake of experimental simplification, we forego the use of a propeller as we focus exclusively on the unpowered climb phase[26]. The experiments were conducted in a room equipped with motion-tracking cameras (Fig. 4a) guaranteeing millimeter-level accurate state estimation (details of estimation and control integration in Supplementary Fig. 3 and Supplementary Note 3). The drone was positioned on an electronically-controlled launcher that accelerated the drone to a speed of $11.6 \pm 0.3 \, ms^{-1}$ into the transition state obtained from the optimal trajectory (further details in "Methods"). A protective net was positioned at the perching distance to ensure the safe recovery of the drone.

Preliminary tests showed that time delays, minor model mismatches, and external disturbances can lead to deviations from the drone trajectory that we precomputed in simulation, necessitating a controller to guide the drone's flight path and ensure adherence to the trajectory. Smith Compensation[32] takes into account time delays and the Nonlinear Model Predictive Control (NMPC) plans actuator commands from the current state of the drone (position, velocity, orientation, angular velocity, and estimated actuator positions) to follow the pre-computed trajectory and correct deviations from the longitudinal direction. This approach enables us to account for actuator limits and address the nonlinear dynamics encountered during the climb phase (further elaborated in Methods). Discrepancies along the lateral direction of motion are compensated with gain-scheduled Proportional-Derivative control of asymmetric wing twist for roll and of tail yaw angle for yaw compensation (see "Methods").

Across twelve consecutive launches (Fig. 4b, Supplementary Movie 5), the drone consistently replicated the climbing maneuver observed in simulation experiments and matched the flight trajectories of 537 flights of similarly sized Harris' hawks that started at a distance of 12 m from the perching target[26].

The total distance covered during the climb phase of the LisEagle can be measured by aligning the flown trajectories with the optimal trajectory to obtain the distance to the impact point assumed at its peak. While the Harris' hawks and simulated drone cover a similar distance (4.2 m and 4.1 m, respectively), the LisEagle covers a slightly longer distance (4.6 m) and impacts at a slightly higher altitude (see Discussion). Regarding the horizontal distance required, the

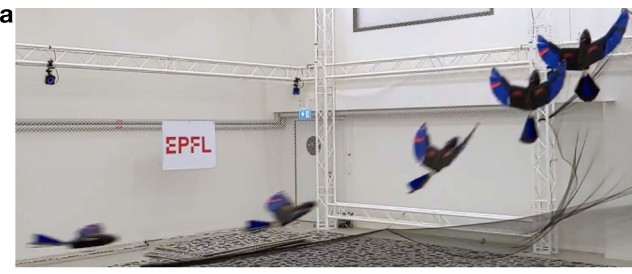

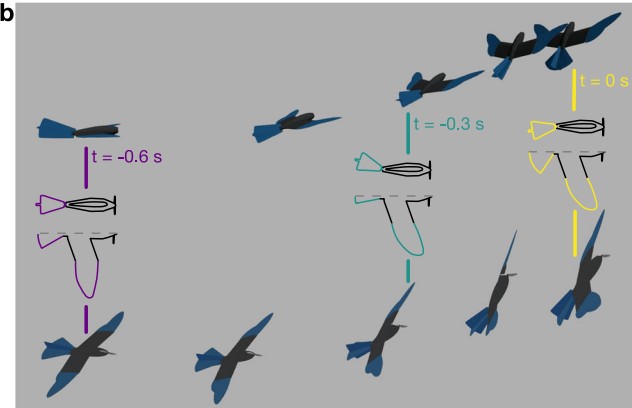

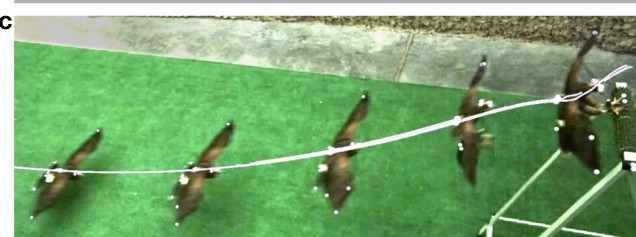

**Fig. 5 | Qualitative comparison of the perching maneuver of the drone, simulation, and bird. a** Perched flight trajectory of the drone. **b** Simulation of the optimal trajectory highlighting the three configurations of the climb phase (also shown with the same color scheme in Fig. 3d). On the top, the sequence is shown observed from a virtual camera at an angle matching the camera's angle used to record the drone's flight trajectory and at the bottom, from a virtual camera at an angle matching (**c**) the Harris' hawk flight recording[26]. Each overlayed image in the sequence was captured 0.15 s apart.

simulation and mean of the LisEagle flights are within the distribution of Harris' hawk flights. In fact, they are less than one standard deviation from the mean required horizontal distance of the bird's flights (0.75 and −0.03 standard deviations away from the Harris' hawk mean, respectively, commonly referred to as z-score, see Supplementary Note 4 and Supplementary Fig. 4 for details).

The simulation indicates that only 18.4% of the kinetic energy is retained. The twelve LisEagle flights display a similar behavior with an average of 12.5% ± 0.7% of kinetic energy retained. Harris' hawks' flights display 14.3% ± 5.1% retention of kinetic energy that closely matches data of simulated flights (z-score of −0.56) and drone flights (z-score of −0.33).

## Discussion

Although the trajectory generation method used optimization criteria derived from bird studies, it did not explicitly aim at imitating flight paths or actuation sequences displayed by birds[12,13,26]. The similarity of the resulting drone trajectories and actuation strategies with those displayed by birds during the agile climb phase, along with the similar distribution of kinetic energy components, strongly corroborates the hypothesis that birds of prey aim at reducing kinetic energy and distance flown at high angles of attack prior to impact with a perching point. These results are further strengthened by the similarity in the distance covered by birds and drones during the climb phase when

minimizing the distance flown at high angles of attack, but not when minimizing the total flight time or required propulsion energy. In addition, our results show that active morphing of wing and tail plays an important role in the climb phase of the perching maneuver. Although drones that could not modify the surface of the wing and tail could produce a perching trajectory, they required a ~50% longer distance in the climb phase and impacted the perching point with almost double the kinetic energy ratio. While we discovered that both tail and wing sweep contribute to enabling this behavior in the drone, performance degradation is more significantly impacted by limitations in wing morphing. Finally, qualitative observations of the morphing sequence of the real drone in the climb phase display a striking resemblance to the actuation sequence and timing of the Harris' hawks filmed in laboratory experiments (Fig. 5).

Although the resulting trajectories and control strategies were validated also on a real drone with similar mass and size to the Harris' hawks, those results revealed small trajectory discrepancies with the simulated drone. While discrepancies due to electro-mechanical imperfections could be characterized and compensated for, aerodynamic phenomena that are exacerbated at high angular rates and accelerations, such as dynamic stall[33], pitch damping, and aeroelastic effects[34], are difficult to accurately model[35] and measure (see Methods). We speculate that these remaining differences could be addressed by data-driven models. For example, approaches such as Gaussian Processes[36] applied to quadrotor dynamics models have been shown to capture aerodynamic effects that appear during agile maneuvers from sparse data and have also been recently used for aerodynamic modeling of a morphing-wing drone that can transition from hovering to forward flight[37].

The kinetic energy at impact on the physical drone closely mirrors that of birds (12.5% LisEagle, 14.3% Harris' hawk). We believe that the slightly higher value observed in simulation experiments (18.4%) can be accredited to the aforementioned modeling limitation and underscore the need for validation on the physical drone.

Overall, the results described here suggest that avian-inspired drones can bridge the gap between observation and systematic analysis of animal behaviors, and serve as an additional tool to unravel the complex interplay of biomechanics, aerodynamics, and control strategies underpinning birds' agile flight behaviors. Notably, this approach can offer a pragmatic alternative to live bird experiments. It simplifies experimental design by reducing the number of experiments that are required by inter-individual differences of complex animals like birds[26,38]. Furthermore, robotic models enable experimental manipulations that could be impossible or unethical in animal studies, such as the locking of articulations that were used here to study the role of wing morphing in the perching maneuver. The method described here could also be applied to other types of perching maneuvers that birds and drones may use, such as initiating the maneuver from different altitudes relative to the perching point (for preliminary results, see Supplementary Fig. 5 and Supplementary Note 5).

We note, however, that while our drone possesses similar physical characteristics in terms of wing area and weight, differences do exist. The wing span of our drone is ~1.5 times larger and, for ethical considerations, the drone is equipped with synthetic feathers instead of bird feathers. These artificial feathers differ from natural feathers in that they lack the interlocking mechanism, potentially resulting in a decrease in lift efficiency[18]. Furthermore, this study focused on the 2.5-dimensional trajectory space of the perching maneuver and was restricted to the optimal control space of wing sweep, tail sweep, and tail incidence. Birds of prey also utilize their alula−a small feathered structure on the leading edge of the wing − as a high-lift device to enhance control in challenging flight regimes, such as post-stall and deep-stall situations[25]. They employ tail rotation[12,39] and adjust wing camber to control roll while maintaining lift[1]. While birds eventually grasp onto the perching surface[31], this mechanical action was not the focus of our study. The method described here could, however, be

extended to this final mechanical phase of a perch by means of a drone with grasping appendages[40] and employing other relevant morphological features in order to test other hypotheses and shed new light on agile flight in 3-dimensional space.

Lastly, the method described could also be used to generate control strategies for a new generation of adaptive flying machines that display optimal agility and energetic efficiency for predefined environmental conditions, such as flying over large distances in windy conditions or maneuvering through confined spaces.

## Methods

### Dynamics model

We developed the drone's aerodynamics model building on the theory described in[41], which uses non-dimensionalized aerodynamic coefficients that are based on stability and control derivatives. We expand on this idea to capture the increased number of degrees of freedom and obtain the derivatives at the expected operation range as described in[9] with a measurement setup consisting of an open-jet wind tunnel (WindShape, Switzerland) that provides a 1.92 m × 1.68 m × 1.5 m test volume and sub−1% turbulence intensity, with a wind speed set at 10 ms$^{-1}$ (Reynolds number: 146,396). The drone is mounted on an ATI Nano25 force and torque balance, interfaced with a National Instruments NI-DAQmx 9.5.1 logger recording at 1 kHz and averaging over five samples. This setup is fixed to the end of a 0.7 m long cylindrical steel tube (external diameter: 0.025 m, internal diameter: 0.02 m), which is connected to a Stäubli TX-90 robotic arm that ensures central positioning in front of the wind tunnel. For the static aerodynamic coefficients, the robotic arm systematically varies the drone's angle of attack (in steps of 4° in the range of [−8°, 40°] and steps of 10° in the range of [40°, 90°]) relative to the wind tunnel flow. We applied the forced oscillation method, where controlled oscillations are applied to study dynamic aerodynamic coefficients[42–44] at −8°, 0°, 12°, and 24°. Here, the range of −8° to 12° angles of attack is intended to capture the angles of attack commonly adopted by birds and winged drones, whereas 24° is chosen to capture the post-stall angle of attack adopted for a short time during the perching maneuver (we observed little change of dynamic coefficients for larger angles of attack of our drone). The forced oscillation experiments are performed at rates of 30° s$^{-1}$ and 60° s$^{-1}$ with oscillation amplitudes of ±4°. While these rates are lower than those typically encountered in flight, the assumption of a linear proportional relationship between dynamic coefficients and angular rates, enabled a close replication of the Harris hawk's flight performance during a perching maneuver. The flexibility of the drone's wings led to strong aeroelastic effects at angular rates higher than 60° s$^{-1}$ leading to noisy dynamic coefficients data. Consequently, we extrapolated our findings by assuming linear proportional relationships between dynamic coefficients and angular rates for our experimental conditions[44]. The robotic arm could maintain a maximum angle error of less than 0.1°, while the drone's actuator positions are adjusted to three discrete positions (min, central, and max position of the values provided in Fig. 2) for tail and wing sweep and tail pitch for each static and dynamic wind tunnel experiment. The tail deflects more upwards than downwards. To accurately capture the aerodynamic forces and moments when the tail is in a straight position, aligned with the body, we take additional measurements under these conditions. Following the procedure of ref. 17, we obtain non-dimensional coefficients for the aerodynamic effects at each angle of attack, angular rate, and for all actuator positions, allowing us to calculate the aerodynamic forces and moment through linear interpolation. Considering the angle of attack $\alpha = \tan(w/u)$, the upward pitch rate $q$, and the normalized actuator positions $\mathbf{a} = [a_{ws}, a_{ts}, a_{ti}, a_{th}]$ ($a_{ws}$ indicates wing sweep, $a_{ts}$ tail sweep, $a_{ti}$ tail incidence, and $a_{th}$ thrust, each parameter normalized between 0 and 1) we employ linear interpolation to obtain the non-dimensional coefficients for the lift $C_L(\alpha, q, \mathbf{a})$, drag $C_D(\alpha, q, \mathbf{a})$, and pitch moment $C_m(\alpha, q, \mathbf{a})$ (visualized in Supplementary Fig. 7). Given

the drone speed $V = \sqrt{u^2 + v^2}$, the coefficients allow the calculation of lift force $f_L$, drag force $f_D$, and pitch moment $m_y$.

$$f_L = \frac{1}{2} \rho V^2 S C_L(\alpha, q, \mathbf{a}) \tag{1}$$

$$f_D = \frac{1}{2} \rho V^2 S C_D(\alpha, q, \mathbf{a}) \tag{2}$$

$$m_y = \frac{1}{2} \rho V^2 S c C_m(\alpha, q, \mathbf{a}) \tag{3}$$

Since these forces can be rotated to be body frame and $\alpha$ and $V$ are functions of $u$ and $w$, we can summarize the aerodynamic forces in the forward $f_x$ and downward $f_z$ direction with respect to the body and the moment about the $y$-axis as $m_y$ as

$$[f_x, f_z, m_y]^\top = \mathbf{f}_{\mathrm{aero}}(u, w, q, \mathbf{a}) \tag{4}$$

We define the system dynamics as a 2-dimensional single-body system where the external force and moment are defined by the aerodynamics model and gravity. Additionally, we assume quasi-static inertia properties. Specifically, we take the shift in center of mass and change of moment of inertia caused by sweeping the wings into account. Data for the shift in center of gravity and moment of inertia at minimum, maximum, and central wing positions are extracted from a CAD model. Linear interpolation of these values at each time step provides a quasi-static approximation for our system dynamics model, reflecting their change due to wing sweep.

### Trajectory optimization method

As we focus on longitudinal motion of the drone, we define the trajectory state $\mathbf{x}_{\mathrm{traj}}$ and trajectory input $\mathbf{u}_{\mathrm{traj}}$

$$\mathbf{x}_{\mathrm{traj}} = [x, z, u, w, \theta, q]^\top \tag{5}$$

$$\mathbf{u}_{\mathrm{traj}} = \mathbf{a} \tag{6}$$

where $x, z$ are the forward and downward position in the world frame, $u, w$ forward and downward velocity in the body frame, $\theta$ the upward pitch angle, and $q$ is the upward pitch rate, following conventions in fixed-wing literature[41].

The system is discretized at times $t_k = \Delta t \cdot k$, such that the drone states $\mathbf{x}_{\mathrm{traj},k}$ are defined at nodes $k \in [0, N]$ and the inputs $\mathbf{u}_{\mathrm{traj},k}$ at nodes $k \in [0, N-1]$, with $N = 100$. We define the trajectory optimization problem using multiple shooting, therefore the decision variables $\mathbf{X}$ contain $\mathbf{x}_{\mathrm{traj},k}$ and $\mathbf{u}_{\mathrm{traj},k}$ of the complete trajectory. Furthermore, $\mathbf{X}$ also contains $\Delta t$ to allow variable time discretization, since the duration of the trajectory is not a priori fixed.

We formulate the trajectory optimization problem based on[45] to find the trajectory $\mathbf{X}^*$ minimizing the cost function $\mathbf{L}(\mathbf{X})$ for $\mathbf{X} \in \mathbb{R}^n$, subject to equality constraints $\mathbf{g}(\mathbf{X})$ and inequality constraints $\mathbf{h}(\mathbf{X})$ as

$$\mathbf{X}^* = \underset{\mathbf{X}}{\arg\min} \ \mathbf{L}(\mathbf{X}) \tag{7}$$

$$\text{subject to } \mathbf{g}(\mathbf{X}) = 0 \ \& \ \mathbf{h}(\mathbf{X}) \leq 0 \tag{8}$$

The equality constraints $\mathbf{g}(\mathbf{X})$ contain the system dynamics, initial conditions, and final constraints of the system. We enforce the system dynamics through constraints

$$\mathbf{x}_{\mathrm{traj},k+1} = \mathbf{x}_{\mathrm{traj},k} + \Delta t \cdot \mathbf{f}_{\mathrm{RK4}}(\mathbf{x}_{\mathrm{traj},k}, \mathbf{u}_{\mathrm{traj},k}) \tag{9}$$

where $\mathbf{f}_{\mathrm{RK4}}(\mathbf{x}_{\mathrm{traj},k}, \mathbf{u}_{\mathrm{traj},k})$ is the fourth-order Runge-Kutta (RK4) integration method of the nonlinear system dynamics. The initial state is defined by a horizontally aligned body with initial forward velocity $\mathbf{x}_{\mathrm{traj},0} = [d,0,u_0,0,0,0]^\top$, where $d$ is the horizontal distance to the target and the initial forward velocity $u_0$, while the initial control input is defined by the central position of the actuators $\mathbf{u}_{\mathrm{traj},0} = [0.5, 0.5, 0.5, 0.5]^\top$. The final state is only constrained by the target position such that the drone trajectory passes through the target point $[0, 0]^\top$.

The inequality constraints $\mathbf{h}(\mathbf{X})$ contain four parts: the actuator limitations, a positivity constraint on time, and the maximal vertical displacement. The actuator limitations constrain the range of motion, actuator velocity, and acceleration according to measurements we conducted in a motion-tracking system while sending step commands to the actuators. The time positivity constraint enforces $\Delta t$ to be positive. Finally, we limit the vertical displacement to maximally 1 m downward from its starting point, similar to observations in the study on Harris' hawks[26] and described in Results.

In the study on Harris' hawks, the final speed is enforced to be equal to that observed in birds in order to define the kinetic energy on impact. However, as the final kinetic energy of the drone is unknown a priori, we include a cost on the final kinetic energy $\mathbf{L}_{E,N}$ alongside the objective described in the study, namely the distance traveled at a high angle of attack $\tilde{\mathbf{L}}_{\alpha,k}$, in the cost function $\mathbf{L}(\mathbf{X})$ as a weighted sum as

$$\mathbf{L}(\mathbf{X}) = w_E\,\mathbf{L}_{E,N} + w_\alpha \sum_{k=1}^{N} \tilde{\mathbf{L}}_{\alpha,k} \tag{10}$$

We first set $w_E$ and then increased $w_\alpha$ by a factor of ten up until we observed that the drone did not reach the stall angle anymore, resulting in $w_E = 0.0063$ and $w_\alpha = 0.01$. We implement the trajectory optimization using CasADi[46] with the Ipopt solver[30], which is a nonlinear solver that is widely employed for trajectory optimization. As Ipopt requires $\mathbf{L}(\mathbf{X})$ to be twice continuously differentiable we enforce a smooth cost function, where

$$\tilde{\mathbf{L}}_{\alpha,k} = V_k \Delta t \cdot \left( \frac{1}{2} \tanh\left( \frac{\alpha_k - \alpha_{\mathrm{stall}}}{\epsilon} \right) + \frac{1}{2} \right) \tag{11}$$

$$\mathbf{L}_{E,N} = \frac{1}{2} m V_N^2 \tag{12}$$

where $V$ denotes the speed and $\tilde{\mathbf{L}}_{\alpha,k}$ is a differentiable approximation of a step function with value $V_k \Delta t$ if $\alpha_k > \alpha_{\mathrm{stall}}$. We choose $\epsilon = 2°$ and define the angle of attack at step $k$ as $\alpha_k = w_k/u_k$ and the stall angle of attack $\alpha_{\mathrm{stall}} = 16°$, which we obtain from the drop-off in the lift curve with actuators at the center position[47].

The implementation in CasADi allows us to compute a trajectory of Fig. 3b within 6.65 s ± 0.40 s on an Apple M2 processor, which is an eight-core CPU clocked at 2.42-3.48 GHz.

### Trajectory tracking of the physical drone

Agile trajectory tracking control of an avian-inspired drone poses three main challenges. Firstly, actuator limitations play an important role since the system dynamics are fast relative to the actuation dynamics, i.e., the system has a rapid transient response, requiring them alongside delays to be considered in the control system. Secondly, while a large amount of control systems assume linear system dynamics, agile flight requires high pitch rates and flight at high angles of attack causing highly nonlinear aerodynamic effects[47]. Lastly, the actuation is coupled, for example, the effect of tail incidence on the drone's pitch rate also depends on the tail sweep. We tackle these challenges with a model-aware control approach using Smith Compensation (SC) and NMPC, allowing us to compensate delays (with SC) and consider

actuation limits while regulating highly nonlinear system dynamics (with NMPC).

Modeling the actuators by limiting their acceleration, velocity, and position was sufficient in the trajectory optimization. However, on the real drone, we need to account for the discrepancy between the desired actuator position that is sent to the servo and the true actuator position. As a result, a more detailed actuator model is needed, which we design as a PD-controlled second-order system with limited acceleration and velocity (further details in Supplementary Note 6). This necessitate incorporation of $\mathbf{a}$ and $\dot{\mathbf{a}}$ to the system state $\mathbf{x}_{\mathrm{mpc}}$, allowing us to define the control inputs $\mathbf{u}_{\mathrm{mpc}}$ as the desired actuator positions $\bar{\mathbf{a}}$, such that

$$\mathbf{x}_{\mathrm{mpc}} = [x, z, u, w, \theta, q, \mathbf{a}^\top, \dot{\mathbf{a}}^\top]^\top \tag{13}$$

$$\mathbf{u}_{\mathrm{mpc}} = \bar{\mathbf{a}} \tag{14}$$

We used the recorded data from actuator step functions to determine the model parameters. To address the delay compensation, we measure the compound transmission and actuation delay and add the NMPC computation delay, resulting in a total delay $\Delta t_{\mathrm{delay}}$ of 0.09 s. To compensate for this delay, we rely on Smith Compensation, i.e., starting from the currently measured state we simulate the system forward in time using the delayed control inputs (details in Supplementary Note 7). This simulation is carried out for a duration of $\Delta t_{\mathrm{delay}}$, resulting in a prediction of the system state $\mathbf{x}_{\mathrm{mpc,pred}}$ at the time when the control inputs will be affecting the system.

We reduce delays by minimizing the computation time of the NMPC by formulating the trajectory tracking problem as a Sequential Quadratic Program (SQP) and utilizing acados, a high-performance embedded optimal control problem solver[48]. We define the trajectory tracking SQP for the avian-inspired drone as

$$\mathbf{Y}^* = \underset{\mathbf{Y}}{\arg\min}\ \mathbf{L}_{\mathrm{mpc}}(\mathbf{Y}) \tag{15}$$

$$\text{subject to } \mathbf{g}_{\mathrm{mpc}}(\mathbf{Y}) = 0\ \&\ \mathbf{h}_{\mathrm{mpc}}(\mathbf{Y}) \le 0 \tag{16}$$

Where $\mathbf{Y}$ contains the system state $\mathbf{x}_{\mathrm{mpc}}$ over the horizon of $M = 20$ steps.

The equality constraints defined by $\mathbf{g}_{\mathrm{mpc}}(\mathbf{Y})$ enforce the system dynamics described above and set the initial condition, such that $\mathbf{x}_0 = \mathbf{x}_{\mathrm{mpc,pred}}$. The inequality constraints $\mathbf{h}_{\mathrm{mpc}}(\mathbf{Y})$ contain only the previously mentioned actuator limitations. The cost is defined as the deviation from the precomputed trajectory $\mathbf{X}^*$, which we obtained by first finding the point closest to $\mathbf{x}_{\mathrm{mpc,pred}}$, from where we sample $M$ points along the trajectory at 30 Hz. We then define the trajectory following cost as

$$\mathbf{J}(\mathbf{Y}) = \sum_{k=1}^{M} \|\mathbf{x}_k - \mathbf{x}_{\mathrm{traj},k}\|_{\mathbf{Q}} \tag{17}$$

where $\mathbf{Q}$ is the state cost weight matrix $\mathbf{Q} = \mathrm{diag}(1000, 1000, 10, 10, 10, 1) \times 10^5$ that weighs the importance of position $x, y$, velocity $u, w$, pitch $\theta$, and pitch rate $q$, obtained through tuning.

### Lateral stabilization of the physical drone

The physical drone can display small lateral trajectory deviations due to lateral asymmetries of the wing and tail hardware. To ensure lateral stabilization of the physical drone, we employ gain scheduled Proportional-Derivative (PD) controllers to stabilize roll and yaw with asymmetric wing twisting and tail yaw deflection respectively. Since aerodynamic moments scale $\propto V^2$, we schedule the PD gains for both controllers with $1/V^2$ if $V \ge 0.5\,\mathrm{ms}^{-1}$ and set them to their central

position otherwise. Using the Ziegler-Nichols method[49] to tune the controller, the resulting gains were $K_{p,roll} = 240$, $K_{d,roll} = 30$, $K_{p,yaw} = 219$, $K_{d,roll} = 38$, assuming normalized actuator output between [0, 1] for actuation ranges shown in Results.

### Drone launcher

To ensure comparable initial speed, position, and attitude over multiple trials before the climb phase, the drone is deployed by a custom-built launcher (design inspired from ref. 40, details in Supplementary Fig. 6, Supplementary Note 8). Over a 1.8 m distance, the launcher propels the drone into horizontal flight. With a constant acceleration, the launcher speeds the drone up at 37.4 ms$^{-2}$, equating to 3.8 g forces, to a final speed of $11.6 \pm 0.3$ ms$^{-1}$. Upon reaching the end of the launcher, the drone is passively released and flies in a horizontal flight condition with an upward pitch angle of $2.9 \pm 0.4°$.

### Reporting summary

Further information on research design is available in the Nature Portfolio Reporting Summary linked to this article.

## Data availability

The experimental raw data generated in this study have been deposited on Zenodo under accession code 10283445[50] and is also available in the Supplementary Data 1. Source data are provided with this paper.

## Code availability

The code that supports the findings of this study and allows the creation of the figures from the raw data can be obtained on Zenodo under accession code 10283445[50] and is also available in the Supplementary Data 1.

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

## Acknowledgements

We thank Roc Arandes, Hauke Maak, and Hoang Vu Phan for help in building and repairing the LisEagle during preliminary tests of the experimental setup. We also thank Raphael Zufferey for inputs on the launcher design; Mohammad Askari for help in the launcher construction; and Charbel Tourmieh for discussions and proofreading the methods. This work was partly supported by the National Centre of Competence in Research (NCCR) Robotics, funded by the Swiss National Science Foundation (grant number 51NF40_185543).

## Author contributions

V.W. and D.F. developed the concept, experimental design, and wrote the manuscript with contributions from all other authors. E.A. developed the drone and characterization method with contributions from M.F. V.W., S.J., and E.A. developed the dynamics model. V.W. and F.B. developed the trajectory optimization method. V.W. and S.J. performed data analysis with contributions by M.F. and D.F. V.W. developed the tracking control methods, developed software, and carried out hardware experiments with contributions from S.J.

## Competing interests

The authors declare no competing interests.
