## [Peer Review File · Nature Communications]

Agile perching maneuvers in birds and morphing-wing dronesREVIEWER COMMENTS

Reviewer #1 (Remarks to the Author):

This study utilizes an avian-inspired drone to validate a recent hypothesis regarding trajectory optimization for perching, which was previously investigated by biological flight tests. Overall, the paper is easy to follow. However, there are still some issues that need to be clarified before getting published.

Concerns:

In lines 222-225, the wing sweep may also cause a forward shift for the center of mass, which can reduce the longitudinal instability effects caused by the forward shift of the neutral point. Please clarify and provide the spatial position of the neutral point and the center of mass.

In Fig. 3c, which shows the perching simulation for the fixed-wing configuration, why didn't the authors try to reduce the distance between the launch position and the perch position, instead of stopping the simulation at -13 meters?

In Equation (1), please elaborate and plot the aerodynamic coefficients matrix used in the dynamic model.

In lines 550-551, it would be beneficial to provide clarification regarding the angular rate (or frequency) of the forced oscillation experiments to test the dynamic aerodynamic coefficient. Whether it is similar to the operating conditions being investigated?

The sentence in line #555-557 is not clear and needs to be rephrased.

In Supplementary Fig. 2, it would be helpful to clarify whether the optimization process, aimed at minimizing time and power, took into consideration a cost associated with the final kinetic energy, L_E , N , as presented in Equation (7).

In Supplementary Note 3, I would suggest the authors create a diagram illustrating the trajectory tracking control framework. This diagram should include components such as feedback, data fusion, PD-based lateral control, MPC-based trajectory tracking control, and their respective control frequencies.

Reviewer #2 (Remarks to the Author):

The authors performed a combination of simulations and experiments on the perching behaviour of a drone with variable tail sweep and wing morphing. Birds display remarkable perching capabilities and this study provides a strong explanation for the observed postural adjustments and trajectories of eagles prior to perching. The work is interesting and in many ways validates several logical deductions.

While the authors have commended job of combining experiments and simulations, it is not clear if the entire parameter space was explored. Can the authors clarify if tests were performed where only the wing was morphed while the tail was not and vice versa? While I appreciate the authors combination of simulations and experiments, it would be more persuasive if the different variables were more rigorously evaluated to highlight the performance manifold the birds are likely operating in.

How were the range of wing morphing chosen in the simulations and experiments? given that the biological measurements were not taken in this study I would have hoped that here too a large range was investigated in the simulations and experiments (within practically reasonable).

Given that there was no actual grasping mechanism or a target location, can the authors also clarify what the controller in the drone did? i.e. was the entire flight sequence preprogrammed? What was the feedback used by the controller in the maneuver? In the absence of this information, it is a bit difficult to build confidence in the generality of the results.

Indeed, the objective of birds would be to dissipate the KE and extending the wings and flaring the tail would achieve this but how are these parameters related in terms of the dynamics of the system?

I am sorry for requesting you to perform more simulations and experiments and provide a more comprehensive analysis of the behaviors. I hope you can understand my point of view. In the absence of these additional analyses I believe the study in the current form might be more suited for a more specialized journal.

Reviewer #3 (Remarks to the Author):

Summary: The authors test the hypothesis that birds minimize distance flown at a high angle of attack while perching using optimization methods in simulations of an avian-inspired drone. The authors then compare simulation results with experimental drone flights and a Harris's Hawk perching study. The results suggest that the morphing of the wing and tail showcased in simulation, drone flight experiments, and avian perching experiments lead to optimal flight trajectories during the perching sequence.

The authors present a method for bridging the gap between simulation and experiments and show results that corroborate a hypothesis and agree across multiple testing mediums. The method is well-documented and based on a strong theoretical background, and the need for optimal control strategies is highly motivated. I believe the article is a valuable addition to the body of literature. However, there are areas that can be addressed to enhance the communication of the motivation and results.

1. The authors should include a definition of agility before stating that the climb is the agile maneuver; it would put the aerodynamic impacts in the right context and would also answer the question of why a dive may not be an "agile" maneuver in this case

2. Page 2, the last paragraph that details the motivation for the study:

- Three motivating points are introduced: anatomical and kinematic interplay, aerodynamic impacts, and optimal control strategies
- The organization of this could be adjusted so that there is a small explanation as to why studying anatomy/kinematic interplay is significant and then a brief summary of the two state-of-the-art experiments presented, making sure to add overall conclusions and gaps.
- The motivating organization for aerodynamic impacts lacks a clear "gap". Maybe a statement suggesting that the authors seek to expose a similar aerodynamic impact in the form of changes in impact energy can be helpful

3. Page 3, second paragraph addressing control strategy motivation

- I would suggest returning to the motivating sentence about optimal control strategies before diving into the SOTA summary. If this is included, the final sentence which states, "The importance of such anticipatory control..." becomes much stronger

4. Page 3, last paragraph discussing what the paper achieves

- It would be good to return again to those three motivating points, stating that anatomical and kinematic interplay is included via the comparison to flight experiments, both biological and man-made, aerodynamic impacts are studied within the simulation methods, and optimal control strategies are achieved by studying flight trajectories (that part is already included, which is good)

5. Page 4, Figure 1

- Highlighting the wrist and elbow joints in both the Harris's Hawk and LisEagle images would be helpful if a connection is to be made between the anatomical and kinematic interplay of these mechanisms; perhaps a skeletal image of the bird wing overlaid with the actual bird, or just markers that indicate where the joints are; this would also help put the "optimal control strategies highlighted in light brown" in more context

6. Page 5, Figure 2

- Refer again to the diving/climbing colors in the figure for the audience
- I suggest a clearer "origin" line for the wing and tail in Figure 2b – for example, is the 130 degrees from the line indicated by the 45 degrees or by the grayish line from which the 45 degrees originate? It's a bit unclear

7. Figure 6, second paragraph detailing the wind tunnel experiments

- Justify the metrics used in comparison in the context of filling the motivation gaps; flight path characterization is necessary for the optimal control strategies I gather, impact energy for aerodynamic impacts, distance for the corroboration with the hypothesis, wing and tail actuation for anatomical/kinematic interplay, etc. This would help the audience understand why this part of the method is so important

8. Page 6, third paragraph, which details trajectory optimization

- It would help the audience to reinforce why the Harris's hawk/Steppe eagle study is being used as a basis here, refer back to that part of the motivation.

9. Page 7, Figure 3

- Consider expanding the size of the figure
- Define "emerging behavior" – what does this mean? Is it referred to later? If not, I don't think this term communicates what you are showing; perhaps "morphing region" or "wing and tail actuation region."
- It looks like in Fig 3c that tail incidence can still change; why? The authors state "without using avian features" but this is still a feature of avian flight; this statement should be more clear and justified

10. Page 7-8, lines 321-327

- This part of the paragraph needs to be adjusted to make it extremely clear to the audience how the impact distance was measured. As it is written, I do not know what the authors mean without having to examine the supplementary files; I should be able to get a relatively clear idea of this measurement without having to look there.
- "results of the bird's posture and leg extension were analyzed when the bird made contact with the perching pole. Considering the absence of leg extension in the drone, energy measurements were conducted with the drone positioned 0.5...."

11. Page 9, line 397

- What is the impact of the drone reducing its tail size? Please clarify

12. Page 10, line 437

- There is a significant error in the kinetic energy retention of the Harris hawk study; is this from estimation? Please justify the statement about a "close match".

13. Page 11, Figure 5

- This figure is not as valuable as a qualitative comparison if the camera angles from the simulation, drone, and hawk study are all different. As a reader, I can't tell that the morphing actuation is even close to similar here, especially between the drone and hawk study. Please consider adjusting the camera angles of the sim and the drone experiment to match the hawk study.

- This should be done for the figure and supplementary video

14. Page 12, second paragraph detailing differences in physical characteristics

- Are aeroelastic effects mentioned? I am presuming that these effects are not incorporated in the

simulation nor on the drone, so it should be mentioned as a significant physical difference

- There is mention of the alula but no mention of whether it could be included beyond the model being extended. I would suggest being specific, stating that actuators could even be added to LisEagle if possible.

15. Page 12, Dynamics model paragraph

- Why are the angles chosen as they are, especially for the dynamic angles? Is it coming from optimization or from estimates from the biological study?

16. Minor comments:

- Page 2, line 050, "do so by" should be replaced by "can", and "adjusting" should be replaced by "adjust the"
- Page 2, line 056, "wing elbow and wrist" should be replaced by "elbow and wrist joints in the wing"
- Page 2, line 068, ", although" should be replaced by "; however,"
- Page 2, line 071, delete "current"
- Page 2, line 075, capitalize the S in Steppe eagle
- Page 2, line 087, replace "by expanding the wings" with "via wing expansion", delete "revealing"
- Page 2, line 088, replace "by folding them as birds do" by "via wing folding, similarly to birds" and add "Additionally," before "Researchers"
- Page 3, line 095, replace "and showed" by ". It was shown"
- Page 3, line 129, replace ", as" with ". Birds"—the sentence should read "Birds commonly do this for migration and commuting and this is also done in robotic..."
- Page 4, line 172, replace "flying machines with morphing wings" with "non-biological flight mechanisms" or something more technical
- Page 5, line 222, is "increases lifting surfaces" always clear? From the figure, I get the sense that the surfaces could either increase or decrease, maybe change to "changes lifting surfaces"
- Page 6, line 276, delete "directing our focus to this critical"
- Page 7, line 311, delete "phase" – the motivation given by the climbing being the agile phase is enough
- Page 10, line 422, delete "instead"
- Page 12, line 510, delete "eliminating the need for breeding birds of prey and can reduce" and replace it with "reducing"
- Page 12, line 516, delete "As such,"
- Page 12, line 537, add in a brief sentence about what the theory details; the reader should be able to get a small sense of the method without having to read the reference

Response to reviews on *Agile perching maneuvers in birds and morphing-wing drones*

V. Wüest, S. Jeger, M. Feroskhan, E. Ajanic, F. Bergonti, and D. Floreano

June 1, 2024

Here we describe the modifications made to the article entitled *Agile Perching Maneuvers in Birds and Morphing-Wing Drones* according to the reviewers' feedback. We express our gratitude to the reviewers for their insightful and predominantly positive comments. We are encouraged by their assessment that the article is easy to follow and well-documented (R1,3), grounded in a solid theoretical framework with a strongly motivated methodology (R3). Additionally, we appreciate the reviewers' comment that our work is both interesting and a valuable contribution to the existing body of literature (R2,3), and that our findings provide a robust explanation for the remarkable perching maneuvers of birds, validating the logical premises of bird observation studies (R2,3).

The highly valuable feedback from the reviewers led us to conduct further simulation experiments investigating the effect of the range of motion of wing and tail sweep on the performance of a perching maneuver (R2,3). We extended the explanation of our aerodynamic model by visualizing the non-dimensional aerodynamic parameters (R1), elaborating on the way we obtained dynamic aerodynamic parameters (R1,3), and analyzing how the center of gravity and aerodynamic center are shifted when sweeping the wings (R1). We also provide a deeper statistical analysis of the similarity between simulation, drone, and bird flights (R3). We addressed multiple ambiguities in the original manuscript (R1,2,3) and detailed the components and interactions between the components involved in the control of our drone (R1,2).

Please note, if not mentioned otherwise, in our responses, we refer to Figures, Tables, Notes, and Movies of the updated manuscript and updated Supplementary Information.

Comments by Reviewer 1

This study utilizes an avian-inspired drone to validate a recent hypothesis regarding trajectory optimization for perching, which was previously investigated by biological flight tests. Overall, the paper is easy to follow. However, there are still some issues that need to be clarified before getting published.

We highly appreciate your insightful suggestions and are committed to clarifying the remaining issues before publication.

In lines 222-225, the wing sweep may also cause a forward shift for the center of mass, which can reduce the longitudinal instability effects caused by the forward shift of the neutral point. Please clarify and provide the spatial position of the neutral point and the center of mass.

Thank you for noticing this important fact. We provide the relevant values in Rebuttal Table 1. In the manuscript, we state that the Neutral Point (NP) shifts forward (positive values) when extending the wing sweep. As you pointed out, the Center of Gravity (CoG) also shifts forward, which reduces the relative position of CoG and NP and thus also needs to be taken into account. We already accounted

for this in our model (see paragraph following Equation 1) and now added Supplementary Note 8 describing these findings, added the table as Supplementary Table 1, and adjusted the manuscript text to reflect this:

“Sweeping the wings forward also changes the position of the center of lift with respect to the center of gravity [14] (collapsed: -1.4 cm, extended: 7.2 cm, see Supplementary Table 1, Supplementary Note 8), thus reducing longitudinal stability and increasing agility.”

configuration	wing sweep angle [°]	CoG position [cm]	NP position [cm]	relative position [cm]
collapsed	45.0	-1.3	-2.7	-1.4
central	87.5	0.0	1.6	1.6
extended	130.0	1.2	8.4	7.2

Rebuttal Table 1: Center of Gravity (CoG) and Neutral Point (NP) position given wing sweep in its collapsed, central, and extended state. We also report their relative position, as it provides crucial insight into the drone’s stability characteristics. Here, we have set the other actuators at their central position and focused solely on wing sweep.

In Fig. 3c, which shows the perching simulation for the fixed-wing configuration, why didn’t the authors try to reduce the distance between the launch position and the perch position, instead of stopping the simulation at -13 meters?

Indeed, we run also simulations starting at positions closer to the target than -13m. For all our experiments, we enforce an initial forward velocity of 10 ms^{-1} and the final position to be the target point through optimization constraints. Given these constraints, in a fixed-wing configuration, trajectories starting from a closer distance than those starting from -13m do not allow sufficient space for the drone to swoop down to -1 m. As a result, these trajectories would not accurately reflect the necessary horizontal distance required for an optimal perching maneuver. Furthermore, they would also not reflect the comparative energy redistribution during a maneuver as observed in bird experiments. We therefore disregarded these flights. We added a sentence in the manuscript to explain our procedure and reasoning:

“Lastly, we excluded trajectories that do not utilize the full 1-meter vertical space, as these would not accurately reflect the necessary horizontal distance required for an optimal perching maneuver or the comparative energy redistribution.”

Thank you for addressing this aspect of the experimental conditions.

In Equation (1), please elaborate and plot the aerodynamic coefficients matrix used in the dynamic model.

Thank you for pointing out this lack of information. In the revised manuscript, we elaborate on the aerodynamic coefficients matrix in multiple ways to address this gap. Firstly, we visualize the aerodynamic coefficient matrix in Rebuttal Figure 1, which we also provide in the Supplementary Information as Supplementary Fig. 5. Secondly, we explain the aerodynamic coefficients more explicitly in the manuscript which now reads:

“[...] we employ linear interpolation to obtain the non-dimensional coefficients for the lift $C_L(\alpha, q, \mathbf{a})$, drag $C_D(\alpha, q, \mathbf{a})$, and pitch moment $C_m(\alpha, q, \mathbf{a})$ (visualized in Supplementary Figure 5). Given the drone speed $V = \sqrt{u^2 + v^2}$, the coefficients allow the calculation of lift force f_L , drag force f_D , and

pitch moment m_y .

$$f_L = \frac{1}{2} \rho V^2 S C_L(\alpha, q, \mathbf{a}) \quad (1)$$

$$f_D = \frac{1}{2} \rho V^2 S C_D(\alpha, q, \mathbf{a}) \quad (2)$$

$$m_y = \frac{1}{2} \rho V^2 S c C_m(\alpha, q, \mathbf{a}) \quad (3)$$

Since these forces can be rotated to the body frame and α and V are functions of u and w , we can summarize the aerodynamic forces in the forward f_x and downward f_z direction with respect to the body and the moment about the y -axis m_y as

$$[f_x, f_z, m_y]^\top = \mathbf{f}_{\text{aero}}(u, w, q, \mathbf{a}) \quad (4)$$

Rebuttal Figure 1: Representation of a selection of the non-dimensional aerodynamic coefficients. Representation of values within the coefficient matrix showing lift coefficients in the left column, drag coefficients in the middle column, and pitch coefficients in the right column. Each row represents the changes of the coefficients along a degree of freedom, where the wing sweep changes are on the top row, tail incidence changes in the middle row, and tail sweep in the bottom row. In each cell, the plot colors represent the actuator positions at min, central, and max position, as described in Methods.

In lines 550-551, it would be beneficial to provide clarification regarding the angular rate (or frequency) of the forced oscillation experiments to test the dynamic aerodynamic coefficient. Whether it is similar to the operating conditions being investigated?

To characterize the dynamic lift coefficient and pitch damping coefficient, we conducted forced pitch oscillation experiments at $30 \text{ }^\circ\text{s}^{-1}$ and $60 \text{ }^\circ\text{s}^{-1}$, following the methodologies akin to those in [22]. Consistent with established practice in literature, we normalized these coefficients and computed the pitch damping moment by multiplying them with the angular rate. We assumed a linear relationship between the

pitch damping coefficient and pitch rate for the entire pitch rate range. The maximum angular rate of $60\text{ }^\circ\text{s}^{-1}$ was set to prevent strong aeroelastic deformations, which led to noisy and unreliable force measurements. During our flight experiments, we later measured pitch angular velocities of up to $297\text{ }^\circ\text{s}^{-1}$. We acknowledge that the angular rates of our forced oscillation measurements were below the angular rates of the flight experiment. We believe this may be a source of the simulation to reality gap we observed. We mentioned this in the Methods and Discussion. We further clarified this shortcoming of our study as follows:

Methods: “We applied the forced oscillation method, where controlled oscillations are applied to study dynamic aerodynamic coefficients [11, 23, 22] at -8° , 0° , 12° , and 24° . Here, -8° to 12° angles of attack represent common flight angles of attack for birds and drones, while 24° represents the high angle of attack regime, beyond which we measured little change in dynamic coefficients. The forced oscillation experiments are performed at rates of $30\text{ }^\circ\text{s}^{-1}$ and $60\text{ }^\circ\text{s}^{-1}$ with oscillation amplitudes of $\pm 4^\circ$. While these rates are lower than those typically encountered in flight, the assumption of a linear proportional relationship between dynamic coefficients and angular rates, enabled a close replication of the Harris’ hawk’s flight performance during a perching maneuver. The flexibility of the drone’s wings led to strong aeroelastic effects at angular rates greater than $60\text{ }^\circ\text{s}^{-1}$ leading to noisy dynamic coefficients data. Consequently, we extrapolated our findings by assuming linear proportional relationships between dynamic coefficients and angular rates for our experimental conditions [22].”

Discussion: “While discrepancies due to electro-mechanical imperfections could be characterized and compensated for, aerodynamic phenomena that are exacerbated at high angular rates and accelerations, such as dynamic stall [15], pitch damping, and aeroelastic effects [9], are difficult to accurately model [4] and measure (see Methods). We speculate that these remaining differences could be addressed by data-driven models.”

The sentence in line #555-557 is not clear and needs to be rephrased.

We agree that this sentence was not easy to understand. We have clarified it to now read:

“The tail deflects more upwards than downwards. To accurately capture the aerodynamic forces and moments when the tail is in a straight position, aligned with the body, we take additional measurements under these conditions.”

In Supplementary Fig. 2, it would be helpful to clarify whether the optimization process, aimed at minimizing time and power, took into consideration a cost associated with the final kinetic energy, L_E , N , as presented in Equation (7).

Thank you for noticing. Indeed, each of the three criteria are minimized alongside the final kinetic energy. We have clarified both the caption of Supplementary Fig. 2 and the corresponding Supplementary Note 2, which state now:

Caption of Supplementary Fig. 2: “Perching flight trajectories resulting from three different optimization objectives, which are minimized alongside the final kinetic energy, as described in Methods.”

Supplementary Note 2: “For each experiment, we apply the same optimization procedure and experimental settings described in the Methods section of the manuscript, including the minimization of the final kinetic energy, as described in Equation 7 of the main text.”

In Supplementary Note 3, I would suggest the authors create a diagram illustrating the trajectory tracking control framework. This diagram should include components such as feedback, data fusion, PD-based lateral control, MPC-based trajectory tracking control, and their respective control frequencies.

We agree that the text describing the control sequence can be difficult to understand. Thank you for the detailed suggestion. We have created a control schematic showing the elements you suggested in Rebuttal Figure 2, added it as Supplementary Fig. 3, and referred to it in the corresponding Supplementary Note 3.

Rebuttal Figure 2: Control system overview. The ground-station computer sends measurements to the onboard Pixhawk 4 autopilot, which fuses the information with inertial measurements to estimate the drone state. The estimated state is passed on to the onboard Nvidia Jetson Nano to control lateral motion with Porportional-Derivative (PD) control and longitudinal motion with trajectory tracking control based on Model Predictive Control (MPC). Finally, both controllers on the Nvidia Jetson Nano send the desired actuator positions to the autopilot.

Comments by Reviewer 2

The authors performed a combination of simulations and experiments on the perching behaviour of a drone with variable tail sweep and wing morphing. Birds display remarkable perching capabilities and this study provides a strong explanation for the observed postural adjustments and trajectories of eagles prior to perching. The work is interesting and in many ways validates several logical deductions.

We thank the reviewer for the positive feedback. Your feedback was tremendously useful and we thank you for your efforts that have strengthened our findings and conclusions further.

While the authors have commendable job of combining experiments and simulations, it is not clear if the entire parameter space was explored. Can the authors clarify if tests were performed where only the wing was morphed while the tail was not and vice versa? While I appreciate the authors combination of simulations and experiments, it would be more persuasive if the different variables were more rigorously evaluated to highlight the performance manifold the birds are likely operating in.

We were intrigued by this remark, thank you for the suggestion. We put additional effort into explaining the performance metrics (ratio of kinetic energy upon impact and required horizontal distance for the climb) as a function of the wing and tail morphing capabilities. We thus conducted additional simulation experiments, where we gradually and independently restricted the range of motion of the wing and tail sweep from their respective central positions. We described the procedure and new findings in the manuscript in Section 2.3, in the Discussion, and we illustrated them in Figure 3c (see Rebuttal Figure 3). In summary, we find that there is a gradual decrease in both performance metrics

when restricting either wing or tail sweep. Notably, we observe a larger performance reduction when restricting wing sweep than when restricting tail sweep.

Rebuttal Figure 3: Effect of a limited range of motion in wing and tail sweep on flight path and key characteristic metrics. The central panel shows the resulting flight paths, the left panel (E kinetic) displays the relative kinetic energy on impact, and the right panel (distance required) indicates the horizontal distance required for the perching maneuver (both surfaces linearly interpolated between experiments, indicated by black dots).

How were the range of wing morphing chosen in the simulations and experiments? given that the biological measurements were not taken in this study I would have hoped that here too a large range was investigated in the simulations and experiments (within practically reasonable).

Thank you for your comment. The range of wing and tail motion was inspired by birds of prey, which in cruise flight reduce their wing and tail size to reduce drag, while during agile maneuvers fully extend wing and tail to increase lift and the control moments. We have elaborated on this in [2, 3] and adjusted the manuscript accordingly:

“The drone used in this study [...] is endowed with feathered wings and tail with controllable sweep and incidence angles observed in birds of prey during agile maneuvers [...] [2, 3].”

To demonstrate how other morphologies affect the kinetic energy and the required horizontal distance, we have added experiments where we test the effects of a limited range of motion of wing and tail sweep. Please see the answer to your previous comment including Rebuttal Figure 3, Section 2.3, Discussion, and Figure 3c for details.

Given that there was no actual grasping mechanism or a target location, can the authors also clarify what the controller in the drone did? i.e. was the entire flight sequence preprogrammed? What was the feedback used by the controller in the maneuver? In the absence of this information, it is a bit difficult to build confidence in the generality of the results.

We acknowledge the need for a better description of these points. It is true that there is no grasping mechanism. The trajectory (flight path and attitudes) of the drone body is precomputed in simulation. We then use Nonlinear Model Predictive control on the drone to plan actuator commands from the current state of the drone (position, velocity, orientation, angular velocity, and estimated actuator positions) to follow the precomputed trajectory. Lastly, we compensate for actuator delays with Smith Compensation.

So, while the trajectory is precomputed, we run closed-loop control on the robot body states. We have adjusted the manuscript to state this more explicitly and refer the reader to the Methods section, where we describe the control method in detail:

“Smith Compensation [1] takes into account time delays and the Nonlinear Model Predictive Control (NMPC) plans actuator commands from the current state of the drone (position, velocity, orientation, angular velocity, and estimated actuator positions) to follow the pre-computed trajectory and correct for deviations from the longitudinal direction. This approach enables us to account for actuator limits and address the nonlinear dynamics encountered during the climb phase (further elaborated in Methods).”

Furthermore, we described the control system integration in Supplementary Note 3 and created Supplementary Figure 3, which illustrates the components of the control system. We hope that these explanations clarify the ambiguities of the original manuscript.

Indeed, the objective of birds would be to dissipate the KE and extending the wings and flaring the tail would achieve this but how are these parameters related in terms of the dynamics of the system?

To understand the role of the parameters of wing sweeping and tail sweeping on the dissipated KE, we have extended our study. We independently and jointly restricted the range of motion of wing and tail sweep and evaluated the resulting impact on the kinetic energy on impact. Please refer to our answer to your second remark regarding the evaluation of the performance manifold, to Section 2.3, to the Discussion, and Figure 3c, where we describe the results in more detail. We hope this answers your question.

I am sorry for requesting you to perform more simulations and experiments and provide a more comprehensive analysis of the behaviors. I hope you can understand my point of view. In the absence of these additional analyses I believe the study in the current form might be more suited for a more specialized journal.

Thank you for your feedback, which led us to better investigate the role of wing and tail morphing. We believe the new results further strengthen our earlier findings and highlight the value of robotic models in studying avian behavior.

Comments by Reviewer 3

Summary: The authors test the hypothesis that birds minimize distance flown at a high angle of attack while perching using optimization methods in simulations of an avian-inspired drone. The authors then compare simulation results with experimental drone flights and a Harris’s Hawk perching study. The results suggest that the morphing of the wing and tail showcased in simulation, drone flight experiments, and avian perching experiments lead to optimal flight trajectories during the perching sequence. The authors present a method for bridging the gap between simulation and experiments and show results that corroborate a hypothesis and agree across multiple testing mediums. The method is well-documented and based on a strong theoretical background, and the need for optimal control strategies is highly motivated. I believe the article is a valuable addition to the body of literature. However, there are areas that can be addressed to enhance the communication of the motivation and results.

Thank you for the overall positive and valuable feedback. We highly appreciate the acknowledgment of the article’s value and are happy to receive such helpful improvement suggestions.

1. The authors should include a definition of agility before stating that the climb is the agile maneuver; it would put the aerodynamic impacts in the right context and would also answer the question of why a dive may not be an "agile" maneuver in this case

This is a great point. We added the definition of agility early in the manuscript by writing "This ability to rapidly change orientation and flight direction epitomizes agility, defined in our context as the ability to rapidly change linear and angular velocities [2]."

2. Page 2, the last paragraph that details the motivation for the study:

- Three motivating points are introduced: anatomical and kinematic interplay, aerodynamic impacts, and optimal control strategies
- The organization of this could be adjusted so that there is a small explanation as to why studying anatomy/kinematic interplay is significant and then a brief summary of the two state-of-the-art experiments presented, making sure to add overall conclusions and gaps.
- The motivating organization for aerodynamic impacts lacks a clear "gap". Maybe a statement suggesting that the authors seek to expose a similar aerodynamic impact in the form of changes in impact energy can be helpful

Thank you for raising the point that part of this section was unclear. There may have been some confusion because the text of this section was ambiguous. You may have understood that robots can be used to investigate (1) the anatomical and kinematic interplay, (2) the aerodynamic impact of wing and tail morphing, and (3) the optimal control strategies independently. Instead, we meant to emphasize that avian-inspired robots have the potential to look at their interplay, i.e. the interplay between (1) anatomical and kinematic configurations, (2) aerodynamic impact of wing and tail morphing, and (3) optimal control strategies. We acknowledge that the sentence was ambiguous and therefore rephrased it to make the causal relation between them more clear:

"Avian-inspired robots can serve as models to systematically investigate the three factors of anatomical and kinematic motion ability of wing and tail morphing, their aerodynamic impact on static and dynamic flight conditions, and the optimal control strategies under a variety of well-defined flight behaviors"

We have also rephrased the beginning of the next paragraph. In this part, we emphasize that in current works, the interplay of these three factors was neglected because actuation and control were simplified. To highlight the need to investigate this interplay, we point to birds who achieve maneuvers like a perched landing by exhibiting complex strategies that would not be possible to execute with current systems. We also mention that the birds' complex actuation patterns are required to dissipate a large amount of kinetic energy:

"Birds exhibit complex control strategies that require the consideration of the interplay of the three previously mentioned factors to balance costs and benefits by swiftly changing morphological configurations. The drones described above, on the other hand, have been steered through comparatively simple maneuvers by either manual teleoperation [8, 2, 3, 7] or reactive scheduled Proportional-Integral (PI) controllers [14]. To fully investigate and leverage the potential of morphing wing and tail during more complex and agile avian-like flight maneuvers, it becomes crucial to achieve high angular acceleration and high aerodynamic forces [20]. However, these demands require opposing actuator inputs. For example, producing high angular acceleration would demand the tail to be inclined upwards, whereas producing a high lift force would require it to be deflected downwards [2]. Therefore, an understanding of the underlying control objective and foresight into upcoming aerodynamic conditions are important to reconcile these conflicting aerodynamic needs. The importance of such anticipatory control strategies becomes even more pronounced when operating near the robot's actuation limits where the actuation bandwidth is severely limited [16, 10].

In this context, the perching maneuver displayed by birds of prey is a remarkable example of an agile

maneuver that demands flight across a diverse aerodynamic spectrum to achieve and balance high angular acceleration and high aerodynamic forces [5], allowing the birds to dissipate kinetic energy within a short distance [6].”

3. Page 3, second paragraph addressing control strategy motivation

- I would suggest returning to the motivating sentence about optimal control strategies before diving into the SOTA summary. If this is included, the final sentence which states, "The importance of such anticipatory control..." becomes much stronger

Thank you for the great point. We added a sentence before the SOTA to set the context of why other control strategies are required. This sets the context properly and emphasizes the last sentence of the paragraph, as you pointed out:

“Birds exhibit complex control strategies, balancing costs and benefits by swiftly changing morphological configurations. The drones described above have been steered through comparatively simple maneuvers by either manual teleoperation [8, 2, 3, 7] or reactive scheduled Proportional-Integral (PI) controllers [14].”

4. Page 3, last paragraph discussing what the paper achieves

- It would be good to return again to those three motivating points, stating that anatomical and kinematic interplay is included via the comparison to flight experiments, both biological and man-made, aerodynamic impacts are studied within the simulation methods, and optimal control strategies are achieved by studying flight trajectories (that part is already included, which is good)

Thank you for suggesting to return to the three points and highlight them more explicitly. We adjusted the paragraph in multiple ways.

Firstly, we adjusted the text to mention that the simulation results reflect the results of the avian study in terms of aerodynamic characteristics of the maneuver:

“The simulation results indicate that the optimal flight path and aerodynamic characteristics of the flown horizontal distance, as well as relative kinetic energy at impact, match those measured in the Harris’ hawks.”

We also added a sentence to capture the fact that we study the impact of morphing in simulation:

“Systematically limiting the range of wing and tail sweep led to a deterioration in aerodynamic performance, emphasizing the importance of the wing kinematics and the anatomical ability to morph.”

5. Page 4, Figure 1

- Highlighting the wrist and elbow joints in both the Harris’s Hawk and LisEagle images would be helpful if a connection is to be made between the anatomical and kinematic interplay of these mechanisms; perhaps a skeletal image of the bird wing overlaid with the actual bird, or just markers that indicate where the joints are; this would also help put the "optimal control strategies highlighted in light brown" in more context

Thank you for your comment. We agree that pointing out the joints in both images may be helpful. Since we did not have access to a Harris’ Hawk skeleton, we had to rely on images that were available online to overlay a skeleton as accurately as possible. We have adapted Figure 1 in the manuscript.

Editorial Note: Figure below redacted where no permission to publish could be obtained.

For completeness, we show in Rebuttal Figure 4 the images that we used to overlay the bird's skeleton and identify the wrist joint and tail feather attachment points.

[REDACTED]

Rebuttal Figure 4: Images used to align the bird's skeleton in Figure 1 in the manuscript. **(a)** The Harris' hawk image, where we overlay the skeleton on the wing and indicate the wrist joint and tail feather attachment points in red. **(b-e)** Images used to help align the skeleton on the wing and identify the wrist joint and tail feather attachment points. **(b)** The skeletal anatomy and feather arrangement of a typical bird wing [13] allowed us to locate the bones with respect to the wing feathers. **(c)** A skeleton of a Harris' hawk [21] that we used to overlay in our figure. **(d)** Image of a Harris' hawk with collapsed tail, which together with **(a)** allows the identification of the position of the tail feather attachment point [17]. **(e)** Close-up view of the tail feather attachment points used to identify relative positions of those points [18].

6. Page 5, Figure 2

- Refer again to the diving/climbing colors in the figure for the audience
- I suggest a clearer "origin" line for the wing and tail in Figure 2b - for example, is the 130 degrees from the line indicated by the 45 degrees or by the grayish line from which the 45 degrees originate? It's a bit unclear

We completely agree with the two comments. Regarding your first point, in the manuscript text, we now refer to the colors used in the figure. Regarding your second point, we agree that it was unclear from where the 130 degrees is measured. We updated the arrows for wing sweep and tail sweep in Figure 2 to clearly capture from where we measure the range of motion angles, as shown in Rebuttal Figure 5.

Rebuttal Figure 5: Illustration of the drone's eight degrees of freedom. This study focuses on longitudinal motion, which is affected by symmetric sweeping of left and right wing, tail incidence, tail sweep, and thrust. The control of these degrees of freedom is given to the trajectory optimization method. Instead, lateral displacements induced by mechanical asymmetries of the drone feathers and actuator responses are stabilized by reactive Proportional-Derivative (PD) controllers where tail yaw correct yaw motion and asymmetric left and right wing twist correct roll motion.

7. Figure 6, second paragraph detailing the wind tunnel experiments

- Justify the metrics used in comparison in the context of filling the motivation gaps; flight path characterization is necessary for the optimal control strategies I gather, impact energy for aerodynamic impacts, distance for the corroboration with the hypothesis, wing and tail actuation for anatomical/kinematic interplay, etc. This would help the audience understand why this part of the method is so important

Thank you for the helpful suggestion. We believe, addressing this comment requires the consideration of the possible misunderstanding mentioned in your comment "2. Page 2, [...]". To address your concern to the best of our ability, we show which metrics we use to evaluate the resulting flight trajectories that the interplay of the three factors ((1) anatomical and kinematic configurations, (2) aerodynamic impact, and (3) optimal control strategies) enables. These metrics and the qualitative comparison of the wing and tail actuation allow us to compare the flight behavior to the Harris' hawk. We added a clear explanation of the used metrics with reasoning. The paragraph now reads:

“We recreate a similar scenario to obtain trajectories that allow a comparative assessment of the resulting control strategies. To compare the performance of our results to the Harris’ hawk, we evaluate the perching performance in terms of relative impact energy (a measure of kinetic energy reduction to the means of reduced impact) and required distance (a measure to compare flight path and space requirements). In Figure 5, we further qualitatively compare the wing and tail actuation sequence leading to the trajectory with the wing and tail motion observed in birds [12, 6, 19].”

8. Page 6, third paragraph, which details trajectory optimization

- It would help the audience to reinforce why the Harris’s hawk/Steppe eagle study is being used as a basis here, refer back to that part of the motivation.

This is a great remark. We added text to reinforce the reasoning behind the use of these studies: “We used trajectory optimization criteria identified in studies of similarly sized birds of prey engaged in the perching maneuver [12, 19, 6] with the goal of testing the hypothesis that birds minimize the distance flown at high angle of attack and of studying the role of morphing strategies in perching trajectories and performance.”

9. Page 7, Figure 3

- Consider expanding the size of the figure
- Define "emerging behavior" - what does this mean? Is it referred to later? If not, I don’t think this term communicates what you are showing; perhaps "morphing region" or "wing and tail actuation region."
- It looks like in Fig 3c that tail incidence can still change; why? The authors state "without using avian features" but this is still a feature of avian flight; this statement should be more clear and justified

We agree that the figure was small and we adjusted the text size. We further note that upon publication, the figure size will be increased by more than 25% (from 14 cm width to 18 cm width).

Thank you for pointing out the lack of explanation of the term “emerging behavior”. We agree that it is ambiguous and replaced it with “agile climb phase” to emphasize that this is the phase where the drone displays high agility.

We have also updated Figure 3c to address your last point. It is indeed true that the drone can still actuate the tail incidence. The experiment aimed at comparing the performance of a configuration that leverages avian morphing features to that of a configuration that is equivalent to a fixed-wing drone. Since fixed-wing drones control longitudinal motion by actuating only the propeller and elevator, we reproduced that configuration by locking wing and tail sweep and allowing only actuation of propeller and tail incidence. If we allowed only the actuation of the propeller, the drone’s control authority would have been inferior to that of a fixed-wing drone. We hope that this explanation answers your question. We have also adjusted the text in Section 2.3 “Wing and tail morphing results in higher kinetic energy dissipation” to reflect this idea more clearly:

“In the condition where wing and tail sweep motions are blocked, only the propeller thrust and tail incidence can control longitudinal motion. This limited configuration most closely resembles that of a fixed-wing drone (Fig. 3c, orange trajectory) and serves as a control condition to understand the contribution of wing and tail sweep to the perching maneuver.”

10. Page 7-8, lines 321-327

- This part of the paragraph needs to be adjusted to make it extremely clear to the audience how the impact distance was measured. As it is written, I do not know what the authors mean without having to examine the supplementary files; I should be able to get a relatively clear idea of this measurement without having to look there.
- "results of the bird's posture and leg extension were analyzed when the bird made contact with the perching pole. Considering the absence of leg extension in the drone, energy measurements were conducted with the drone positioned 0.5...."

We fully agree with the comment. We rephrased the section to explain our method more clearly: "The bird's posture and ability to extend its legs forward enable it to touch the perching pole before the body reaches the pole. We estimated this effective distance at 0.5 m by analyzing video footage of Harris' hawk experiments (detailed in Supplementary Fig. 1, Supplementary Note 1). Considering the absence of leg extension in the drone, energy measurements were conducted when the drone reached a distance of 0.5 m from the perching target."

11. Page 9, line 397

- What is the impact of the drone reducing its tail size? Please clarify

Here we want to highlight that in the second configuration of the climb phase (Figure 3d, green configuration at 0.3s) the trajectory optimization algorithm folds the tail while the bird is reported to expand its tail at this stage. Since we suspected that in our aerodynamic model the collapsed tail configuration produced a higher pitch-up moment than the extended configuration, we measured the aerodynamic pitch-moment with (a) the collapsed tail configuration chosen by the trajectory optimization method and with (b) the extended tail configuration, all other things being equal. We observed that the collapsed tail (a) produced a 0.07 Nm pitch-up moment, while the extended tail (b) produced a -0.55 Nm pitch-down moment. The reduced pitch-up moment of the extended tail is caused by the flight at high angle of attack [2] and increased rotational drag induced by the high angular rate of 140°s^{-1} at this stage of the climbing phase. To capture this info in the manuscript, we adjusted the text to state:

"[...] although, differently from birds, it reduces the tail size, which in our simulation leads to a higher pitch-up moment of 0.07 Nm, than an extended tail -0.55 Nm at this stage of the maneuver".

12. Page 10, line 437

- There is a significant error in the kinetic energy retention of the Harris hawk study; is this from estimation? Please justify the statement about a "close match".

Thank you for raising this point. We would like to clarify our statement regarding the "close match" between the key metrics of the Harris' Hawk compared to the simulation and LisEagle flights. We visualize the distribution of relative kinetic energy on impact and horizontal distance from the transition point to the perch for each condition (bird, drone, simulation) in Rebuttal Figure 6.

The "close match" refers to the fact that the simulation and mean values of the drone flights lie within the distribution of the Harris' hawk flights. To quantify this statement, we further calculate the z-score (deviation of a measurement from the mean, expressed in standard deviations) of the simulation and mean of the drone flights with respect to the distribution of Harris' hawk flights. We find a z-score of -0.56 and -0.03 for the simulation and -0.33 and 0.75 for the drone flights for the kinetic energy

Rebuttal Figure 6: Distribution of key characteristic data of bird, drone, and simulation flights. The figures show the distribution of (a) the relative kinetic energy on impact and (b) the horizontal distance from the transition point to the perch point. We show the histogram of all flights and show an approximation of a Gaussian distribution.

and horizontal distance, respectively. The z-scores thus indicate that all metrics lie within less than a standard deviation from the mean of Harris’ hawk flights. We think that the term “closely matching” is appropriate as it reflects the statistical proximity of the simulation and drone flights to the Harris’ hawk flight metrics.

We have revised the manuscript to explicitly state the z-scores, their significance, and point to a more detailed explanation in Supplementary Note 7 and Supplementary Fig. 6:

“While the Harris’ hawks and simulated drone cover a similar distance (4.2 m and 4.1 m, respectively), the LisEagle covers a slightly longer distance (4.6 m) and impacts at a slightly higher altitude (see Discussion). Regarding the horizontal distance required, the simulation and mean of the LisEagle flights are within the distribution of Harris’ hawk flights. In fact, they are less than one standard deviation from the mean required horizontal distance of the bird’s flights (0.75 and -0.03 standard deviations away from the Harris’ hawk mean, respectively, commonly referred to as z-score, see Supplementary Note 7 and Supplementary Fig. 6 for details).

The simulation indicates that only 18.4% of the kinetic energy is retained. The twelve LisEagle flights display a similar behavior with an average of $12.5\% \pm 0.7\%$ of kinetic energy retained. Harris’ hawks’ flights display $14.3\% \pm 5.1\%$ retention of kinetic energy that closely matches data of simulated flights (z-score of -0.56) and drone flights (z-score of -0.33).”

13. Page 11, Figure 5

- This figure is not as valuable as a qualitative comparison if the camera angles from the simulation, drone, and hawk study are all different. As a reader, I can’t tell that the morphing actuation is even close to similar here, especially between the drone and hawk study. Please consider adjusting the camera angles of the sim and the drone experiment to match the hawk study.
- This should be done for the figure and supplementary video

We had chosen to record at this angle because it is difficult to position a camera in our experimental room at an overhead position similar to that of the avian study. However, we agree with you that this makes the comparison more difficult. Therefore, we adjusted the angle of the virtual camera that renders the flight of the simulated drone twice: once to the footage of the drone flights and once to the footage of the bird flights. This allows comparison between simulation & drone and simulation & bird, and also allows to better show the actuation of the drone’s wing and tail. We adjusted the text and Figure 5 in the manuscript (as shown in Rebuttal Figure 7). Lastly, we also updated the Supplementary Movie 4.

Rebuttal Figure 7: Qualitative comparison of the perching maneuver in simulation, drone flights, and bird flights. (a) Perched flight in simulation highlighting the three configurations of the climb phase (also shown in Fig. 3d) observed from a virtual camera at an angle matching the camera's angle used to record (b) the drone's flight trajectory. (c) Perched flight in simulation with a virtual camera at an angle matching the camera's angle used to record (d) the Harris' hawk flight [12]. Each overlaid image in the sequence was captured 0.15 s apart.

14. Page 12, second paragraph detailing differences in physical characteristics

- Are aeroelastic effects mentioned? I am presuming that these effects are not incorporated in the simulation nor on the drone, so it should be mentioned as a significant physical difference
- There is mention of the alula but no mention of whether it could be included beyond the model being extended. I would suggest being specific, stating that actuators could even be added to LisEagle if possible.

Thank you for your raising these points. We agree that aeroelastic effects may be a partial cause for the observed discrepancies. We had already mentioned this in the Discussion section of the original manuscript:

“While discrepancies due to electro-mechanical imperfections could be characterized and compensated for, aerodynamic phenomena that are exacerbated at high angular rates and accelerations, such as dynamic stall [15], pitch damping, and aeroelastic effects [9], are difficult to accurately model [4] and measure (see Methods). We speculate that these remaining differences could be addressed by data-driven models.”

Regarding the alula, as you mention, the drone could be extended to incorporate this additional degree of freedom to gather additional insights into the role of the alula. We had mentioned this idea in the Discussion section of the original manuscript:

“However, birds of prey also utilize their alula [...]. The method described here could be extended to these and other factors by means of a drone with relevant morphological features [...].”

We therefore left the manuscript unchanged. Please let us know if stronger emphasis or better clarification is necessary.

15. Page 12, Dynamics model paragraph

- Why are the angles chosen as they are, especially for the dynamic angles? Is it coming from optimization or from estimates from the biological study?

Our objective was to understand how the aerodynamic surfaces behave within the range of angles of attack where birds are most commonly observed flying, specifically between -8° and 12° . During perching maneuvers, however, birds operate also in the post-stall regime for a short period of time; this is why we performed measures also at 24° angle of attack. We have revised the Methods section to better explain this:

“Here, the range of -8° to 12° angles of attack is intended to capture the angles of attack commonly adopted by birds and winged drones, whereas 24° is chosen to capture the post-stall angle of attack adopted for a short time during the perching maneuver (we observed little change of dynamic coefficients for larger angles of attack of our drone). The forced oscillation experiments are performed at rates of 30°s^{-1} and 60°s^{-1} with oscillation amplitudes of $\pm 4^\circ$. While these rates are lower than those typically encountered in flight, the assumption of a linear proportional relationship between dynamic coefficients and angular rates, enabled a close replication of the Harris’ hawk’s flight performance during a perching maneuver. The flexibility of the drone’s wings led to strong aeroelastic effects at angular rates higher than 60°s^{-1} leading to noisy dynamic coefficients data. Consequently, we extrapolated our findings by assuming linear proportional relationships between dynamic coefficients and angular rates for our experimental conditions [22].”

16. Minor comments:

- Page 2, line 050, "do so by" should be replaced by "can", and "adjusting" should be replaced by "adjust the"
- Page 2, line 056, "wing elbow and wrist" should be replaced by "elbow and wrist joints in the wing"
- Page 2, line 068, ", although" should be replaced by "; however,"
- Page 2, line 071, delete "current"
- Page 2, line 075, capitalize the S in Steppe eagle
- Page 2, line 087, replace "by expanding the wings" with "via wing expansion", delete "revealing"
- Page 2, line 088, replace "by folding them as birds do" by "via wing folding, similarly to birds" and add "Additionally," before "Researchers"
- Page 3, line 095, replace "and showed" by ". It was shown"
- Page 3, line 129, replace ", as" with ". Birds"|the sentence should read "Birds commonly do this for migration and commuting and this is also done in robotic..."
- Page 4, line 172, replace "flying machines with morphing wings" with "non-biological flight mechanisms" or something more technical
- Page 5, line 222, is "increases lifting surfaces" always clear? From the figure, I get the sense that the surfaces could either increase or decrease, maybe change to "changes lifting surfaces"
- Page 6, line 276, delete "directing our focus to this critical"
- Page 7, line 311, delete "phase" - the motivation given by the climbing being the agile phase is enough
- Page 10, line 422, delete "instead"
- Page 12, line 510, delete "eliminating the need for breeding birds of prey and can reduce" and replace it with "reducing"
- Page 12, line 516, delete "As such,"
- Page 12, line 537, add in a brief sentence about what the theory details; the reader should be able to get a small sense of the method without having to read the reference

Thank you for the detailed suggestions. We highly appreciated them as they improved the flow and readability of the text. We implemented all the suggested changes in the manuscript.

References

- [1] N. Abe and K. Yamanaka. Smith predictor control and internal model control—a tutorial. In *SICE 2003 Annual Conference (IEEE Cat. No. 03TH8734)*, volume 2, pages 1383–1387. IEEE, 2003.
- [2] E. Ajanic, M. Feroskhan, S. Mintchev, F. Noca, and D. Floreano. Bioinspired wing and tail morphing extends drone flight capabilities. *Science Robotics*, 5(47):eabc2897, 2020.
- [3] E. Ajanic, M. Feroskhan, V. Wüest, and D. Floreano. Sharp turning maneuvers with avian-inspired wing and tail morphing. *Communications Engineering*, 1(1):34, 2022.
- [4] S. L. Brunton and B. R. Noack. Closed-loop turbulence control: Progress and challenges. *Applied Mechanics Reviews*, 67(5):050801, 2015.
- [5] A. C. Carruthers, A. L. Thomas, and G. K. Taylor. Automatic aeroelastic devices in the wings of a steppe eagle *aquila nipalensis*. *Journal of Experimental Biology*, 210(23):4136–4149, 2007.
- [6] A. C. Carruthers, A. L. Thomas, S. M. Walker, and G. K. Taylor. Mechanics and aerodynamics of perching manoeuvres in a large bird of prey. *The Aeronautical Journal*, 114(1161):673–680, 2010.
- [7] E. Chang, L. Y. Matloff, A. K. Stowers, and D. Lentink. Soft biohybrid morphing wings with feathers underactuated by wrist and finger motion. *Science Robotics*, 5(38):eaay1246, 2020.
- [8] M. Di Luca, S. Mintchev, G. Heitz, F. Noca, and D. Floreano. Bioinspired morphing wings for extended flight envelope and roll control of small drones. *Interface focus*, 7(1):20160092, 2017.
- [9] E. H. Dowell. *A modern course in aeroelasticity*, volume 217. Springer, 2014.
- [10] P. Foehn, E. Kaufmann, A. Romero, R. Penicka, S. Sun, L. Bauersfeld, T. Laengle, G. Cioffi, Y. Song, A. Loquercio, et al. Agilicious: Open-source and open-hardware agile quadrotor for vision-based flight. *Science robotics*, 7(67):eabl6259, 2022.
- [11] V. Klein and P. Murphy. Estimation of aircraft unsteady aerodynamic parameters from dynamic wind tunnel testing. In *AIAA Atmospheric Flight Mechanics Conference and Exhibit*, page 4016, 2001.
- [12] M. KleinHeerenbrink, L. A. France, C. H. Brighton, and G. K. Taylor. Optimization of avian perching manoeuvres. *Nature*, 607(7917):91–96, 2022.
- [13] D. Liu, B. Song, W. Yang, X. Yang, D. Xue, and X. Lang. A brief review on aerodynamic performance of wingtip slots and research prospect. *Journal of Bionic Engineering*, 18:1255–1279, 2021.
- [14] Y. Liu, J. Zhang, L. Gao, Y. Zhu, B. Liu, X. Zang, H. Cai, and J. Zhao. Employing wing morphing to cooperate aileron deflection improves the rolling agility of drones. *Advanced Intelligent Systems*, page 2300420, 2023.
- [15] K. Mulleners and M. Raffel. Dynamic stall development. *Experiments in fluids*, 54:1–9, 2013.
- [16] Y. Song, A. Romero, M. Müller, V. Koltun, and D. Scaramuzza. Reaching the limit in autonomous racing: Optimal control versus reinforcement learning. *Science Robotics*, 8(82):eadg1462, 2023.
- [17] Taxidermy. Bottom of the tail, 2010. [Online; accessed March 27, 2024].
- [18] The Peregrine Fund. Harris’s hawk, *parabuteo unicinctus*, 2022. [Online; accessed March 27, 2024].
- [19] A. L. Thomas. On the aerodynamics of birds’ tails. *Philosophical Transactions of the Royal Society of London. Series B: Biological Sciences*, 340(1294):361–380, 1993.
- [20] A. L. Thomas. The flight of birds that have wings and a tail: variable geometry expands the envelope of flight performance. *Journal of Theoretical Biology*, 183(3):237–245, 1996.

- [21] Tumblr. Corvus corax, harris hawk, 2021. [Online; accessed March 27, 2024].
- [22] D. D. Vicroy, K. C. Huber, T. D. Loeser, and D. Rohlf. Low-speed dynamic wind tunnel test analysis of a generic 53 swept ucav configuration. In *32nd AIAA Applied Aerodynamics Conference*, page 2003, 2014.
- [23] D. D. Vicroy, T. D. Loeser, and A. Schütte. Static and forced-oscillation tests of a generic unmanned combat air vehicle. *Journal of Aircraft*, 49(6):1558–1583, 2012.

REVIEWERS' COMMENTS

Reviewer #1 (Remarks to the Author):

Most of the issues raised in the previous round have been addressed. Generally, I agree with this response though the revised manuscript skips some concerns. There is still some distance to getting published. To further improve the content of this paper, the authors should consider the following points:

1. The term "perching" may imply the process of contacting the target point until a stable stop is achieved, whereas this paper focuses on the process of a bird or drone approaching the target point.
2. This paper mainly considers situations where the departure and target points have the same or slightly different heights. If scenarios involve a bird or drone diving from a high altitude to the ground and then decelerating to land, or climbing from a low altitude to land on a high rooftop, would the behaviors differ significantly from the situations described in this paper? Considering these scenarios would enhance the universality of the research results.
3. The comparative effect between the bird and the drone in Fig. 1(a) is not ideal. It might be worth adjusting the wing angles to make them look more symmetrical.
4. The content in Fig. 5(a) and (c) is somewhat repetitive. These figures could be merged into a single subfigure.

Reviewer #2 (Remarks to the Author):

The authors have satisfactorily addressed my comments.

Reviewer #3 (Remarks to the Author):

Thank you for carefully addressing all the comments.

Response to reviews on second review round on *Agile perching maneuvers in birds and morphing-wing drones*

V. Wüest, S. Jeger, M. Feroskhan, E. Ajanic, F. Bergonti, and D. Floreano

August 18, 2024

Here, we describe the modifications made to the article entitled *Agile Perching Maneuvers in Birds and Morphing-wing Drones* according to the second round of reviewer feedback. We thank the reviewers for their insightful and predominantly positive comments, which led us to improve Fig. 1 and Fig. 5, clarify the remaining ambiguities, and conduct further simulation experiments.

Comments by Reviewer 1

Most of the issues raised in the previous round have been addressed. Generally, I agree with this response though the revised manuscript skips some concerns. There is still some distance to getting published.

We would like to express our gratitude to the reviewer for their valuable input and constructive comments. We have carefully considered and addressed the remaining concerns to improve the clarity and quality of our manuscript.

To further improve the content of this paper, the authors should consider the following points:

1. The term "perching" may imply the process of contacting the target point until a stable stop is achieved, whereas this paper focuses on the process of a bird or drone approaching the target point.

Thank you for pointing out this potential ambiguity. We understand that the term "perching" may include physically grasping the target until a stable stop is achieved, while the focus of our study is on the approach flight rather than the physical grasp. We find the most closely related avian observation studies [2, 3, 5] and robotics articles [6, 7] speak of a "perching maneuver" when referring exclusively to the approach flight of a perch instead. We therefore clarify this potential ambiguity and consistently refer to the approach flight to a perch as "perching maneuver". We further explain in the introduction that we aim to recreate experiments of avian studies finding that a perching maneuver consists of a dive and a climb phase and specifically state that we exclude the physical grasping. We adjusted the abstract, introduction, and discussion of future work accordingly.

To clarify this potential ambiguity early in the manuscript, we state in the abstract:

"Avian perching *maneuvers* are one of the most frequent and agile flight scenarios, where highly optimized *flight* trajectories [...] reduce kinetic energy at impact."

"Here, we use optimal control methods [...] to test a recent hypothesis derived from perching *maneuvers* of Harris' hawks that birds minimize the distance flown at high angles of attack to dissipate kinetic energy *before impact*."

Multiple sentences in the introduction were altered to reflect this more clearly:

"In this context, the perching maneuver displayed by birds of prey is a remarkable example of an agile

flight maneuver, which is characterized by a dive phase and a climb phase, demanding flight across a diverse aerodynamic spectrum to achieve and balance high angular acceleration and high aerodynamic forces [1]. The perching maneuver allows the birds to dissipate kinetic energy within a short distance in preparation for the impact and grasp [2].

“Here, we test this hypothesis using optimization methods in aerodynamically-grounded simulations of an avian-inspired drone with morphing wing and tail (Fig. 2) to study the *dive and climb phase* of the perching maneuver (*excluding the physical grasping*).”

And we adjusted the discussion to:

“While birds eventually grasp onto the perching surface [4], this mechanical action was not the focus of our study. The method described here could, however, be extended to this final mechanical phase of a perch by means of a drone with grasping appendages [8] [...].

2. This paper mainly considers situations where the departure and target points have the same or slightly different heights. If scenarios involve a bird or drone diving from a high altitude to the ground and then decelerating to land, or climbing from a low altitude to land on a high rooftop, would the behaviors differ significantly from the situations described in this paper? Considering these scenarios would enhance the universality of the research results.

Thank you for your interesting question, which prompted us to perform additional experiments with varying initial altitudes. However, since our article aims at replicating experimental conditions of previous avian studies [2, 3] where the altitude of the starting point did not change, we describe the results of the additional experiments in the Supplementary Information and mention those findings in the discussion section:

“The method described here could also be applied to other types of perching maneuvers that birds and drones may use, such as initiating the maneuver from different altitudes relative to the perching point (for preliminary results, see Supplementary Fig. 5 and Supplementary Note 5).”

In the Supplementary Information, we added the following text and Rebuttal Figure 1 as Supplementary Note 5 and Supplementary Fig. 5:

“Although in this article we aimed at replicating the experimental conditions used in bird studies [2, 3], we also used the method to investigate perching maneuvers starting from different altitudes compared to the perching point, a situation that birds may encounter in nature. Specifically, we used the same conditions described in the manuscript, *i.e.* the drone started in straight flight condition 12 m away from the target at the nominal speed of 10 ms^{-1} , the target was located at 0 m altitude, and the drone was not allowed to touch the ground at -1 m during the dive phase. Given these constraints and the dynamics of our avian-inspired drone, we minimized impact energy and distance flown at high angle of attack for different initial altitudes, namely -0.9, 0, 1, 2, 3, and 4 m relative to the target altitude). The results shown in Supplementary Fig. 5 indicate that the drone could reduce the kinetic energy at impact for all altitude variations. However, starting altitudes higher than 1 m relative to the target point resulted in longer horizontal distance allocated to the dive phase and in longer distance flown at high angles of attack in the climb phase. Whether these maneuvers are adopted also by birds at those altitudes remains to be studied.”

Rebuttal Figure 1: Perching optimization with varying initial altitude. Flight trajectories starting at an initial horizontal distance of 12 m from the target, at initial altitudes of -0.9, 0, 1, 2, 3, and 4 m relative to the target altitude (indicated by the red cross). Graphs show flight altitude, speed, and angle of attack (α) during the dive and climb phase for each trajectory. Circles denote the transition state at the lowest point of the trajectory, which remains consistent for trajectories starting at altitudes between -0.9 m and 1 m (shown in blue) relative to the target. As the initial altitude increases above 1 m relative to the target, the transition point (gray) from dive to climb shifts closer to the target location, the flight speed at the transition increases, and the drone flies at high angles of attack for a longer distance during the climb phase in order to dissipate higher kinetic energy.

3. The comparative effect between the bird and the drone in Fig. 1(a) is not ideal. It might be worth adjusting the wing angles to make them look more symmetrical.

We appreciate your suggestion regarding Fig. 1(a) in the manuscript and have adjusted the viewing angles on the drone to improve the comparative effect and emphasize the symmetry between the bird and the drone. We show the adjusted graphic in Rebuttal Figure 2.

Rebuttal Figure 2: The perching maneuver by Harris' hawks and avian-inspired drone. Schematic overview of a typical perching maneuver consisting of a dive phase (light blue line) and of an agile climb phase (blue line) (adapted from [3], see Supplementary Movie 1); the illustrated bird shows the optimal control strategies displayed by the Harris' hawk (light brown) and by the avian-inspired drone (black/blue).

4. The content in Fig. 5(a) and (c) is somewhat repetitive. These figures could be merged into a single subfigure.

We appreciate your feedback and have taken it into consideration. We have merged the content of Fig. 5(a) and (c) into a single subfigure, as suggested, and adjusted the caption as reflected in Rebuttal Figure 3. We also modified the manuscript accordingly to ensure clarity and consistency.

Rebuttal Figure 3: Qualitative comparison of the perching maneuver in drone flights, simulation, and bird flights. (a) The drone's flight trajectory of the perched flight experiment. (b) Simulation highlighting the three configurations of the climb phase (also shown in Fig. 3d) as viewed from a virtual camera positioned at an angle corresponding to the camera used to record the drone flight at the top, and a virtual camera angle at the bottom, matching the camera's angle used for recording (c) the Harris' hawk flight [3]. Each overlaid image in the sequence was captured 0.15 s apart.

Comments by Reviewer 2

The authors have satisfactorily addressed my comments.

We would like to express our gratitude to the reviewer for their valuable feedback. We appreciate the opportunity to improve our work and thank the reviewer for their contribution to the quality of the article which led to additional insights into the parameter space and various clarifications in terms of control and morphing ranges.

Comments by Reviewer 3

Thank you for carefully addressing all the comments.

We would like to extend our sincere thanks to the reviewer for their meticulous and highly detailed feedback. The reviews allowed us to address ambiguities and clarify statements with statistical analysis that may be important for readers.

Other alterations

Please note that we changed the image of the Harris' hawk in Fig. 1b, Fig. 4c, and Supplementary Fig. 2d due to copyright complications with the originally used photo. We show an overview of the altered parts of the figures in Rebuttal Figure 4.

Rebuttal Figure 4: Overview of the changed figures, showing the relevant section of (a) Fig. 1b, (b) Fig. 4c, and (c) Supplementary Fig. 2d.

Furthermore, we removed all uses of italics and bold font in the main text of the manuscript and re-ordered backmatter sections to comply with the journal requirements.

References

- [1] A. C. Carruthers, A. L. Thomas, and G. K. Taylor. Automatic aeroelastic devices in the wings of a steppe eagle *aquila nipalensis*. *Journal of Experimental Biology*, 210(23):4136–4149, 2007.
- [2] A. C. Carruthers, A. L. Thomas, S. M. Walker, and G. K. Taylor. Mechanics and aerodynamics of perching manoeuvres in a large bird of prey. *The Aeronautical Journal*, 114(1161):673–680, 2010.
- [3] M. KleinHeerenbrink, L. A. France, C. H. Brighton, and G. K. Taylor. Optimization of avian perching manoeuvres. *Nature*, 607(7917):91–96, 2022.
- [4] W. R. Roderick, M. R. Cutkosky, and D. Lentink. Bird-inspired dynamic grasping and perching in arboreal environments. *Science Robotics*, 6(61):eabj7562, 2021.
- [5] A. L. Thomas. On the aerodynamics of birds’ tails. *Philosophical Transactions of the Royal Society of London. Series B: Biological Sciences*, 340(1294):361–380, 1993.
- [6] A. Waldock, C. Greatwood, F. Salama, and T. Richardson. Learning to perform a perched landing on the ground using deep reinforcement learning. *Journal of intelligent & robotic systems*, 92:685–704, 2018.
- [7] A. M. Wickenheiser and E. Garcia. Optimization of perching maneuvers through vehicle morphing. *Journal of Guidance, Control, and Dynamics*, 31(4):815–823, 2008.
- [8] R. Zufferey, J. Tormo-Barbero, D. Feliu-Talegón, S. R. Nekoo, J. Á. Acosta, and A. Ollero. How ornithopters can perch autonomously on a branch. *Nature Communications*, 13(1):7713, 2022.